# META LEARNING LOW RANK COVARIANCE FACTORS FOR ENERGY-BASED DETERMINISTIC UNCERTAINTY

**Jeffrey Ryan Willette[1], Hae Beom Lee[1], Juho Lee[1,2], & Sung Ju Hwang[1,2]**
KAIST[1], AITRICS[2]
`{jwillette,haebeom.lee,juholee,sjhwang82}@kaist.ac.kr`

## ABSTRACT

Numerous recent works utilize bi-Lipschitz regularization of neural network layers to preserve relative distances between data instances in the feature spaces of each layer. This distance sensitivity with respect to the data aids in tasks such as uncertainty calibration and out-of-distribution (OOD) detection. In previous works, features extracted with a distance sensitive model are used to construct feature covariance matrices which are used in deterministic uncertainty estimation or OOD detection. However, in cases where there is a distribution over tasks, these methods result in covariances which are sub-optimal, as they may not leverage all of the meta information which can be shared among tasks. With the use of an attentive set encoder, we propose to meta learn either diagonal or diagonal plus low-rank factors to efficiently construct task specific covariance matrices. Additionally, we propose an inference procedure which utilizes scaled energy to achieve a final predictive distribution which is well calibrated under a distributional dataset shift.

## 1 INTRODUCTION

Accurate uncertainty in predictions (calibration) lies at the heart of being able to trust decisions made by deep neural networks (DNNs). However, DNNs can be miscalibrated when given out-of-distribution (OOD) test examples (Ovadia et al., 2019; Guo et al., 2017). Hein et al. (2019) show that the problem can arise from ReLU non-linearities introducing linear polytopes into decision boundaries which lead to arbitrary high confidence regions outside of the domain of the training data. Another series of works (van Amersfoort et al., 2021; Liu et al., 2020a; Mukhoti et al., 2021; van Amersfoort et al., 2021) link the problem to feature collapse, whereby entire regions of feature space collapse into singularities which then inhibits the ability of a downstream function to differentiate between points in the singularity, thereby destroying any information which could be used to differentiate them. When these collapsed regions include areas of OOD data, the model loses any ability to differentiate between in-distribution (ID) and OOD data.

A solution to prevent feature collapse is to impose bi-Lipschitz regularization into the network, enforcing both an upper and lower Lipschitz bound on each function operating in feature space (van Amersfoort et al., 2021; Liu et al., 2020a), preventing feature collapse. Such features from bi-Lipschitz regualarized extractors are then used to improve downstream tasks such as OOD detection or uncertainty quantification. Broadly speaking, previous works have done this by constructing covariance matrices from the resulting features in order to aid in uncertainty quantification (Liu et al., 2020a; Van Amersfoort et al., 2020) or OOD detection (Mukhoti et al., 2021). Intuitively, features from a Lipschitz regularized extractor make for more expressive covariances, due to the preservation of identifying information within different features.

However, empirical covariance estimation is limited when there are few datapoints on hand, such as in few-shot learning. A key aspect of meta-learning is to learn meta-knowledge over a task distribution, but as we show, empirical covariance estimation methods are not able to effectively encode such knowledge, even when the features used to calculate the covariance come from a meta-learned feature extractor (see Figure 6). As a result, the empirical covariance matrices are not expressive given limited data and thus the model loses its ability to effectively adapt feature covariances to each task.

Another obstacle, highlighted by Mukhoti et al. (2021), is that plain softmax classifiers cannot accurately model epistemic uncertainties. We identify a contributing factor to this, which is the shift

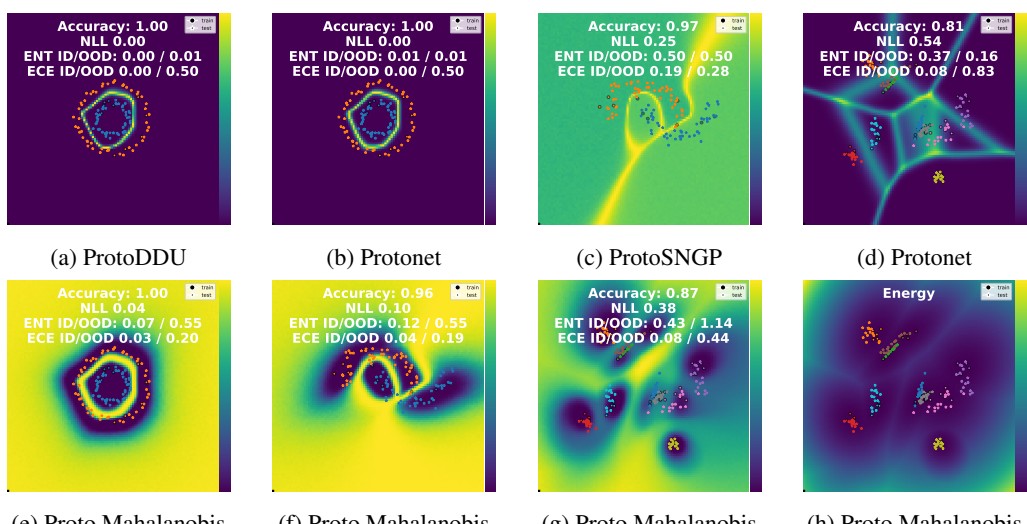

Figure 1: **Top row**: Examples of the learned entropy surface of baseline networks. **Bottom row:** our Proto Mahalanobis models. Each pixel in the the background color represents the entropy given to that coordinate in input space. Baseline networks exhibit high confidence in areas where there has been no evidence, leading to higher calibration error when presented with OOD data.

invariance property of the softmax function. Specifically, even if an evaluation point comes from an OOD area and is assigned low logit values (high energy), this alone is insufficient for a well calibrated prediction. Small variations in logit values can lead to arbitrarily confident predictions due to the shift invariance. From the perspective of Prototypical Networks (Snell et al., 2017), we highlight this problem in Figure 3, although it applies to linear softmax classifiers as well.

In the following work, we first propose a method of meta-learning class-specific covariance matrices that is transferable across the task distribution. Specifically, we meta-learn a function that takes a set of class examples as an input and outputs a class-specific covariance matrix which is in the form of either a diagonal or diagonal plus low-rank factors. By doing so, the resulting covariance matrices remain expressive even with limited amounts of data. Further, in order to tackle the limitation caused by the shift invariance property of the softmax function, we propose to use scaled energy to parameterize a logit-normal softmax distribution which leads to better calibrated softmax scores. We enforce its variance to increase as the minimum energy increases, and vice versa. In this way, the softmax prediction can become progressively more uniform between ID and OOD data, after marginalizing the logit-normal distribution (see example in Figure 1).

By combining those two components, we have an inference procedure which achieves a well calibrated probabilistic model using a deterministic DNN. Our contributions are as follows:

- We show that existing approaches fail to generalize to the meta-learning setting.

- We propose a meta learning framework which predicts diagonal or low-rank covariance factors as a function of a support set.

- We propose an energy-based inference procedure which leads to better calibrated uncertainty on OOD data.

## 2 RELATED WORK

**Mahalanobis Distance.** Mahalanobis distance has been used in previous works for OOD detection (Lee et al., 2018) which also showed that there is a connection between softmax classifiers and Gaussian discriminant analysis, and that the representation space in the latent features of DNN's provides for an effective multivariate Gaussian distribution which can be more useful in constructing class conditional Gaussian distributions than the output space of the softmax classifier. The method outlined in Lee et al. (2018) provides a solid groundwork for our method, which also utilizes

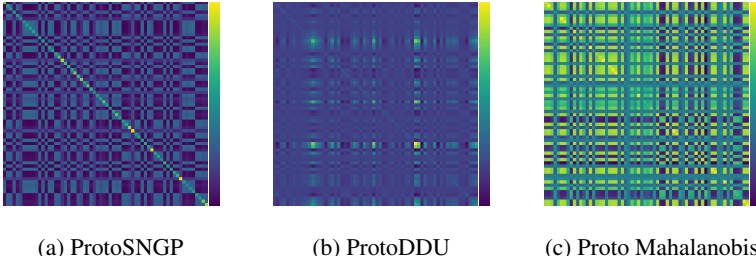

|  (a) ProtoSNGP | (b) ProtoDDU | (c) Proto Mahalanobis |

Figure 2: Comparison between covariances learned in SNGP (Liu et al., 2020a), DDU (Mukhoti et al., 2021) and Proto Mahalanobis (Ours) in the few shot setting (half-moons 2-way/5-shot). Covariance generated from SNGP are close to a multiple of the identity matrix, while that of ProtoMahalanobis contains significant contributions from off-diagonal elements.

Mahalanobis distance in the latent space, and adds a deeper capability to learn meta concepts which can be shared over a distribution of tasks.

**Post Processing.** We refer to post-processing as any method which applies some function after training and before inference in order to improve the test set performance. In the calibration literature, temperature scaling (Guo et al., 2017) is a common and effective post-processing method. As the name suggests, temperature scaling scales the logits by a constant (temperature) before applying the softmax function. The temperature is tuned such that the negative log-likelihood (NLL) on a validation set is minimized. Previous works which utilize covariance (Lee et al., 2018; Mukhoti et al., 2021; Liu et al., 2020a) have also applied post-processing methods to construct latent feature covariance matrices after training. While effective for large single tasks, these post-processing methods make less expressive covariances in the meta learning setting, as demonstrated in Figure 1.

**Bi-Lipschitz Regularization.** Adding a regularizer to enforce functional smoothness of a DNN is a useful tactic in stabilizing the training of generative adversarial networks (GANs) (Miyato et al., 2018; Arjovsky et al., 2017), improving predictive uncertainty (Liu et al., 2020a; Van Amersfoort et al., 2020), and aiding in OOD detection (Mukhoti et al., 2021). By imposing a smoothness constraint on the network, distances which are semantically meaningful w.r.t. the feature manifold can be preserved in the latent representations, allowing for downstream tasks (such as uncertainty estimation) to make use of the preserved information. (Van Amersfoort et al., 2020) showed that without this regularization, a phenomena known as feature collapse can map regions of feature space onto singularities (Huang et al., 2020), where previously distinct features become indistinguishable. For both uncertainty calibration and OOD detection, feature collapse can map OOD features onto the same feature spaces as ID samples, adversely affecting both calibration and OOD separability.

**Meta Learning.** The goal of meta learning (Schmidhuber, 1987; Thrun & Pratt, 1998) is to leverage shared knowledge which may apply across a distribution of tasks. In the few shot learning scenario, models leverage general meta-knowledge gained through episodic training over a task distribution (Vinyals et al., 2016; Ravi & Larochelle, 2017), which allows for effective adaptation and inference on a task which may contain only limited amounts of data during inference. The current meta-learning approaches are roughly categorized into metric-based (Vinyals et al., 2016; Snell et al., 2017) or optimization-based approaches (Finn et al., 2017; Nichol et al., 2018). In this work, our model utilizes a metric-based approach as they are closely related to generative classifiers, which have been shown to be important for epistemic uncertainty (Mukhoti et al., 2021).

## 3 APPROACH

We start by introducing a task distribution $p(\tau)$ which randomly generates tasks containing a support set $\mathcal{S} = \{(\tilde{\mathbf{x}}_i, \tilde{y}_i)\}_{i=1}^{N_s}$ and a query set $\mathcal{Q} = \{(\mathbf{x}_i, y_i)\}_{i=1}^{N_q}$. Then, given randomly sampled task $\tau = (\mathcal{S}, \mathcal{Q})$, we meta-learn a generative classifier that can estimate the class-wise distribution of query examples, $p(\mathbf{x}|y = c, \mathcal{S})$ conditioned on the support set $\mathcal{S}$, for each class $c = 1, \ldots, C$. A generative classifier is a natural choice in our setting due to fact that it utilizes feature space densities which has been shown to be a requirement for accurate epistemic uncertainty prediction (Mukhoti et al., 2021). Under the class-balanced scenario $p(y = 1) = \cdots = p(y = C)$ we can easily predict

the class labels as follows.

$$p(y = c|\mathbf{x}, \mathcal{S}) = \frac{p(\mathbf{x}|y = c, \mathcal{S})}{\sum_{c'=1}^{C} p(\mathbf{x}|y = c', \mathcal{S})}. \tag{1}$$

## 3.1 LIMITATIONS OF EXISTING GENERATIVE CLASSIFIERS

Possibly one of the simplest forms of deep generative classifier is Prototypical Networks (Snell et al., 2017). In Protonets we assume a deep feature extractor $f_\theta$ that embeds $\mathbf{x}$ to a common metric space such that $\mathbf{z} = f_\theta(\mathbf{x})$. We then explicitly model the class-wise distribution $p(\mathbf{z}|y = c, \mathcal{S})$ of the embedding $\mathbf{z}$ instead of the raw input $\mathbf{x}$. Under the assumption of a regular exponential family distribution for $p_\theta(\mathbf{z}|y = c, \mathcal{S})$ and a Bregman divergence $d$ such as Euclidean or Mahalanobis distance, we have $p_\theta(\mathbf{z}|y = c, \mathcal{S}) \propto \exp(-d(\mathbf{z}, \boldsymbol{\mu}_c))$ (Snell et al., 2017), where $\boldsymbol{\mu}_c = \frac{1}{|\mathcal{S}_c|} \sum_{\tilde{\mathbf{x}} \in \mathcal{S}_c} f_\theta(\tilde{\mathbf{x}})$ is the class-wise embedding mean computed from $\mathcal{S}_c$, the set of examples from class $c$. In Protonets, $d$ is squared Euclidean distance, resulting in the following likelihood of the query embedding $\mathbf{z} = f_\theta(\mathbf{x})$ in the form of a softmax function.

$$p_\theta(y = c|\mathbf{z}, \mathcal{S}) = \frac{\exp(-\|\mathbf{z} - \boldsymbol{\mu}_c\|^2)}{\sum_{c'=1}^{C} \exp(-\|\mathbf{z} - \boldsymbol{\mu}_{c'}\|^2)}. \tag{2}$$

**1. Limitations of fixed or empirical covariance.** Unfortunately, Eq. (2) cannot capture a nontrivial class-conditional distribution structure, as Euclidean distance in Eq. (2) is equivalent to Mahalanobis distance with fixed covariance $\mathbf{I}$ for all classes, such that $p_\theta(\mathbf{z}|y = c, \mathcal{S}) = \mathcal{N}(\mathbf{z}; \boldsymbol{\mu}_c, \mathbf{I})$. For this reason, many-shot models such as SNGP (Liu et al., 2020b) and DDU (Mukhoti et al., 2021) calculate empirical covariances from data after training to aid in uncertainty quantification. However, such empirical covariance estimations are limited especially when the dataset size is small. If we consider the few-shot learning scenario where we have only a few training examples for each class, empirical covariances can provide unreliable estimates of the true class covariance. Unreliable covariance leads to poor estimation of Mahalanobis distances and therefore unreliable uncertainty estimation.

**2. Shift invariant property of softmax and OOD calibration.** Another critical limitation of Eq. (2) is that it produces overconfident predictions in areas distant from the class prototypes. The problem can arise from the shift invariance property of the softmax function $\sigma(\omega) = e^\omega / \sum_{\omega'} e^{\omega'}$ with $\omega$ denoting the logits, such

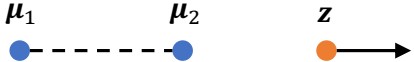

Figure 3: $\|\boldsymbol{\mu}_2 - \boldsymbol{\mu}_1\|$ remains the same while $\mathbf{z}$ travels along the line, making a prediction with unnecessarily low entropy.

that $\sigma(\omega + s) = e^{\omega+s} / \sum_{\omega'} e^{\omega'+s} = e^\omega / \sum_{\omega'} e^{\omega'} = \sigma(\omega)$ for any shift $s$. More specifically, suppose we have two classes $c = 1, 2$, and $\mathbf{z}$ moves along the line extrapolating the prototypes $\boldsymbol{\mu}_1$ and $\boldsymbol{\mu}_2$ such that $\mathbf{z} = \boldsymbol{\mu}_1 + c(\boldsymbol{\mu}_2 - \boldsymbol{\mu}_1)$ for $c \leq 0$ or $c \geq 1$. Then, we can easily derive the following equality based on the shift invariant property of the softmax function:

$$p_\theta(y = 1|\mathbf{z}, \mathcal{S}) = \frac{1}{1 + \exp(\pm\|\boldsymbol{\mu}_2 - \boldsymbol{\mu}_1\|)} \tag{3}$$

where $\pm$ corresponds to the sign of $c$. Note that the expression is invariant to the value of $c$ except for its sign. Therefore, even if $\mathbf{z}$ is OOD, residing somewhere distant from the prototypes $\boldsymbol{\mu}_1$ and $\boldsymbol{\mu}_2$ with extreme values of $c$, we still have equally confident predictions. See Figure 3 for illustration.

## 3.2 META-LEARNING OF THE CLASS-WISE COVARIANCE

In order to remedy the limitations of empirical covariance, and capture a nontrivial structure of the class-conditional distribution even with a small support set, we propose to meta-learn the class-wise covariances over $p(\tau)$. Specifically, we meta-learn a set encoder $g_\phi$ that takes a class set $\mathcal{S}_c$ as input and outputs a covariance matrix corresponding to the density $p(\mathbf{z}|y = c, \mathcal{S})$, for each class $c = 1, \ldots, C$. We expect $g_\phi$ to encode shared meta-knowledge gained through episodic training over tasks from $p(\tau)$, which, as we will demonstrate in section 4, fills a key shortcoming of applying existing methods such as DDU (Mukhoti et al., 2021) and SNGP (Liu et al., 2020a). We denote the set-encoder $g_\phi$ for each class $c$ as

$$\Lambda_c, \Phi_c = g_\phi(\mathcal{Z}_c), \qquad \mathcal{Z}_c = \{\tilde{\mathbf{z}} - \boldsymbol{\mu}_c | \tilde{\mathbf{z}} = f_\theta(\tilde{\mathbf{x}}) \text{ and } \tilde{\mathbf{x}} \in \mathcal{S}_c\}. \tag{4}$$

---

**Algorithm 1** Proto Mahalanobis – Training

1: **Input:** Task distribution $p(\tau)$, initial $\theta$ and $\phi$
2: **Output:** Meta-learned $\theta$ and $\phi$
3: **while** not converged **do**
4:      Sample a task $\tau = (\mathcal{S}, \mathcal{Q})$
5:      **for** $c = 1$ **to** $C$ **do**
6:          $\boldsymbol{\mu}_c \leftarrow \frac{1}{|\mathcal{S}_c|} \sum_{\tilde{\mathbf{x}} \in \mathcal{S}_c} f_\theta(\tilde{\mathbf{x}})$
7:          $\Lambda_c, \Phi_c \leftarrow g_\phi(\mathcal{Z}_c)$       $\triangleright$ Eq. 4
8:          $\boldsymbol{\Sigma}_c \leftarrow \Lambda_c + \Phi_c \Phi_c^\top$       $\triangleright$ Eq. 5
9:          Compute $\boldsymbol{\Sigma}_c^{-1}$ and $|\boldsymbol{\Sigma}_c|$       $\triangleright$ Eq. 8,9
10:      **end for**
11:      $\mathcal{L}_\tau \leftarrow \frac{1}{|\mathcal{Q}|} \sum_{(\mathbf{x},y) \in \mathcal{Q}} - \log p_{\theta,\phi}(y | \mathbb{E}[\boldsymbol{\omega}])$   $\triangleright$ Eq. 10
12:      $(\theta, \phi) \leftarrow (\theta, \phi) - \alpha \nabla_{\theta,\phi} \mathcal{L}_\tau$
13: **end while**

**Algorithm 2** Proto Mahalanobis – Inference

1: **Input:** Task $\tau$, meta-learned $\theta$ and $\phi$
2: **for** $c = 1$ **to** $C$ **do**
3:      $\boldsymbol{\mu}_c \leftarrow \frac{1}{|\mathcal{S}_c|} \sum_{\tilde{\mathbf{x}} \in \mathcal{S}_c} f_\theta(\tilde{\mathbf{x}})$
4:      $\Lambda_c, \Phi_c \leftarrow g_\phi(\mathcal{Z}_c)$       $\triangleright$ Eq. 4
5:      $\boldsymbol{\Sigma}_c \leftarrow \Lambda_c + \Phi_c \Phi_c^\top$       $\triangleright$ Eq. 5
6:      Compute $\boldsymbol{\Sigma}_c^{-1}$ and $|\boldsymbol{\Sigma}_c|$       $\triangleright$ Eq. 8,9
7: **end for**
8: Eval. $p_{\theta,\phi}(y | \mathbf{z}, \mathcal{S})$ for $(y, \mathbf{x}) \in \mathcal{Q}$    $\triangleright$ Eq. 11

---

where $\Lambda_c \in \mathbb{R}^{d \times d}$ is a diagonal matrix and $\Phi_c \in \mathbb{R}^{d \times r}$ is a rank-$r$ matrix. Now, instead of the identity covariance matrix or empirical covariance estimation, we have the meta-learnable covariance matrix consisting of the strictly positive diagonal and low-rank component for each class $c = 1, \ldots, C$.

$$\boldsymbol{\Sigma}_c = \Lambda_c + \Phi_c \Phi_c^\top. \tag{5}$$

It is easy to see that $\boldsymbol{\Sigma}_c$ is a valid positive semi-definite covariance matrix for positive $\Lambda_c$. Note that the covariance becomes diagonal when $r = 0$. A natural choice for $g_\phi$ is the Set Transformer (Lee et al., 2019) which models pairwise interactions between elements of the input set, an implicit requirement for covariance matrices.

Now, we let $p_{\theta,\phi}(\mathbf{z} | y = c, \mathcal{S}) = \mathcal{N}(\mathbf{z}; \boldsymbol{\mu}_c, \boldsymbol{\Sigma}_c)$. From Bayes' rule (see Appendix A.1), we compute the predictive distribution in the form of softmax function as follows,

$$p_{\theta,\phi}(y = c | \mathbf{z}, \mathcal{S}) = \frac{p_{\theta,\phi}(\mathbf{z} | y = c, \mathcal{S})}{\sum_{c'=1}^{C} p_{\theta,\phi}(\mathbf{z} | y = c', \mathcal{S})} \tag{6}$$

$$= \frac{\exp(-\frac{1}{2}(\mathbf{z} - \boldsymbol{\mu}_c)^\top \boldsymbol{\Sigma}_c^{-1}(\mathbf{z} - \boldsymbol{\mu}_c) - \frac{1}{2} \log |\boldsymbol{\Sigma}_c|)}{\sum_{c'=1}^{C} \exp(-\frac{1}{2}(\mathbf{z} - \boldsymbol{\mu}_{c'})^\top \boldsymbol{\Sigma}_{c'}^{-1}(\mathbf{z} - \boldsymbol{\mu}_{c'}) - \frac{1}{2} \log |\boldsymbol{\Sigma}_{c'}|)} \tag{7}$$

**Covariance inversion and log-determinant.** Note that the logit of the softmax function in Eq. (7) involves the inverse covariance $\boldsymbol{\Sigma}_c^{-1}$ and the log-determinant $\log |\boldsymbol{\Sigma}_c|$. In contrast to both DDU and SNGP which propose to calculate and invert an empirical feature covariance during post-processing, the meta-learning setting requires that this inference procedure be performed on every iteration during meta-training, which may be cumbersome if a full $\mathcal{O}(d^3)$ inversion is to be performed. Therefore, we utilize the matrix determinant lemma (Ding & Zhou, 2007) and the Sherman-Morrison formula in the following recursive forms for both the inverse and the log determinant in Equation 7.

$$(\boldsymbol{\Sigma}_i + \Phi_{i+1}\Phi_{i+1}^\top)_{i+1}^{-1} = \boldsymbol{\Sigma}_i^{-1} - \frac{\boldsymbol{\Sigma}_i^{-1}\Phi_{i+1}\Phi_{i+1}^\top\boldsymbol{\Sigma}_i^{-1}}{1 + \Phi_{i+1}^\top\boldsymbol{\Sigma}_i^{-1}\Phi_{i+1}} \tag{8}$$

$$\det(\boldsymbol{\Sigma}_i + \Phi_{i+1}\Phi_{i+1}^\top)_{i+1} = (1 + \Phi_{i+1}^\top\boldsymbol{\Sigma}_i^{-1}\Phi_{i+1})\det(\boldsymbol{\Sigma}_i) \tag{9}$$

## 3.3 OUT-OF-DISTRIBUTION CALIBRATION WITH SCALED ENERGY

Next, in order to tackle the overconfidence problem caused softmax shift invariance (Figure 3), we propose incorporating a positive constrained function of energy $h(E) = \max(\epsilon, -\frac{1}{T} \log \sum_c \exp(-E_c))$, with temperature $T$, into the predictive distribution. Energy has been used for OOD detection (Liu et al., 2020b) and density estimation (Grathwohl et al., 2019), and the success of energy in these tasks implies that it can be used to calibrate the predictive distribution (example in Figure 1h). Results in Grathwohl et al. (2019) show improvements in calibration, but their training procedure requires a full input space generative model during training, adding unwanted complexity if the end goal does not require input space generation. Our method makes use of our logit values $\boldsymbol{\omega} = (\omega_1, \ldots, \omega_C)$ to parameterize the mean of a logit-normal distribution with the variance given by $h(E)$. In this

Table 1: OOD ECE on models trained on variations of the Omniglot and MiniImageNet datasets. The OOD distribution for these models are random classes from the test set which are not present in the support set.

| Model | Omniglot OOD Class ECE ↓ | | | | MiniImageNet OOD Class ECE ↓ | |
| | 5-way 5-shot | 5-way 1-shot | 20-way 5-shot | 20-way 1-shot | 5-way 1-shot | 5-way 5-shot |
| --- | --- | --- | --- | --- | --- | --- |
| MAML | 63.14±0.67 | 53.90±0.77 | 56.60±5.98 | 48.39±1.09 | 29.00±0.67 | 42.43±0.51 |
| Reptile | 48.01±0.76 | 41.84±0.98 | 46.31±0.30 | **35.62±0.49** | 29.86±0.73 | 38.35±0.93 |
| Protonet | 68.50±0.69 | 67.64±0.63 | 77.58±0.37 | 72.07±0.63 | 33.23±1.20 | 47.06±1.30 |
| Protonet-SN | 69.43±0.57 | 67.67±0.70 | 77.84±0.44 | 72.36±0.58 | 33.24±2.14 | 46.76±1.40 |
| ProtoDDU | 69.16±0.63 | 66.61±1.15 | 78.14±0.19 | 71.39±0.74 | 35.31±2.09 | 46.82±1.28 |
| ProtoSNGP | 65.39±0.64 | 60.22±0.61 | 76.90±0.72 | 68.16±0.40 | 34.38±1.21 | 45.84±0.81 |
| Ours (Diag) | 33.95±0.98 | 40.52±0.68 | **40.00±0.23** | 50.39±1.84 | **17.19±1.80** | **32.22±3.12** |
| Ours (Rank 1) | **33.19±0.94** | **39.62±2.02** | 40.04±0.40 | 49.28±1.21 | 18.78±1.72 | 34.44±0.64 |

way, the logit-normal distribution variance rises in conjunction with the energy magnitude, making predictions more uniform over the simplex for higher magnitude energies.

$$p_{\theta,\phi}(\omega_c|\mathbf{z},\mathcal{S}) = \mathcal{N}(\omega_c;\tilde{\mu}_c,\tilde{\sigma}), \quad \text{where} \quad \tilde{\mu}_c = -\frac{1}{2}(\mathbf{z}-\boldsymbol{\mu}_c)^\top \boldsymbol{\Sigma}_c^{-1}(\mathbf{z}-\boldsymbol{\mu}_c) - \frac{1}{2}\log|\boldsymbol{\Sigma}_c|,$$
$$\tilde{\sigma} = -\frac{1}{T}\log\sum_{c'}\exp\left(-(\mathbf{z}-\boldsymbol{\mu}_{c'})^\top \boldsymbol{\Sigma}_{c'}^{-1}(\mathbf{z}-\boldsymbol{\mu}_{c'})\right) \tag{10}$$

Intuitively, $h(E)$ is dominated by $\min_c(|E_c|)$ thereby acting as a soft approximation to the minimum energy magnitude (shortest Mahalanobis distance), which only becomes large when the energy is high for all classes represented in the logits. Then, the predictive distribution becomes

$$p_{\theta,\phi}(y=c|\mathbf{z},\mathcal{S}) = \int p(y=c|\boldsymbol{\omega})p_{\theta,\phi}(\boldsymbol{\omega}|\mathbf{z},\mathcal{S})d\boldsymbol{\omega} \tag{11}$$

$$\approx \frac{1}{M}\sum_{m=1}^{M}\frac{\exp(\omega_c^{(m)})}{\sum_{c'}\exp(\omega_{c'}^{(m)})}, \quad \omega_c^{(m)} \sim p(\omega_c|\mathbf{z},\mathcal{S}). \tag{12}$$

**Meta-training** At training time, we do not sample $\boldsymbol{\omega}$ and use the simple deterministic approximation $p_{\theta,\phi}(y|\mathbf{z},\mathcal{S}) \approx p_{\theta,\phi}(y|\mathbb{E}[\boldsymbol{\omega}])$. Therefore, the loss for each task becomes $\mathcal{L}_\tau(\theta,\phi) = \frac{1}{|\mathcal{Q}|}\sum_{(\mathbf{x},y)\in\mathcal{Q}} -\log p_{\theta,\phi}(y|\mathbb{E}[\boldsymbol{\omega}])$. We then optimize $\theta$ and $\phi$ by minimizing the expected loss $\mathbb{E}_{p(\tau)}[\mathcal{L}_\tau(\theta,\phi)]$ over the task distribution $p(\tau)$ via episodic training.

**Energy scaling.** Inference with equation 11 can still benefit from temperature scaling of $\tilde{\sigma}$ in 10. Therefore, in order properly scale the variance to avoid underconfident ID performance, we tune the temperature parameter $T$ after training. Specifically, we start with $T=1$ and iteratively increase $T$ by 1 until $\mathbb{E}_\mathcal{D}[-\log p(y|\mathbf{z},\mathcal{S})] \leq \mathbb{E}_\mathcal{D}[-\log p(y|\mathbb{E}[\boldsymbol{\omega}])]$, where $-\log p(y|\mathbb{E}[\boldsymbol{\omega}])$ is the NLL evaluated by using only the deterministic logits $\mathbb{E}[\boldsymbol{\omega}]$.

### 3.4 SPECTRAL NORMALIZATION.

Lastly, we enforce a bi-Lipschitz regularization $f_\theta$ by employing both residual connections and spectral normalization on the weights (Liu et al., 2020a), such that Equation 13 is satisfied. Using features $\mathbf{Z}$, the calculation of covariance $(\mathbf{Z}-\boldsymbol{\mu}_c)(\mathbf{Z}-\boldsymbol{\mu}_c)^\top$ and the subsequent mean and variance of 10 both implicitly utilize distance, therefore we require bi-Lipschitz regularization of $f_\theta$. We choose spectral normalization via the power iteration method, also known as the Von Mises Iteration (Mises & Pollaczek-Geiringer, 1929), due to its low memory and computation overhead as compared to second order methods such as gradient penalties (Arjovsky et al., 2017). Specifically, for features at hidden layer $h(\cdot)$, at depth $l$, and for some constants $\alpha_1, \alpha_2$, for all $\mathbf{z}_i$ and $\mathbf{z}_j$, we enforce:

$$\alpha_1||\mathbf{z}_i^{(l)}-\mathbf{z}_j^{(l)}|| \leq ||h(\mathbf{z}_i^{(l-1)})-h(\mathbf{z}_j^{(l-1)})|| \leq \alpha_2||\mathbf{z}_i^{(l)}-\mathbf{z}_j^{(l)}||. \tag{13}$$

## 4 EXPERIMENTS

The goal of our experimental evaluation is to answer the following questions. 1) What is the benefit of each component of our proposed model? 2) Does $g_\phi$ produce more expressive covariances than

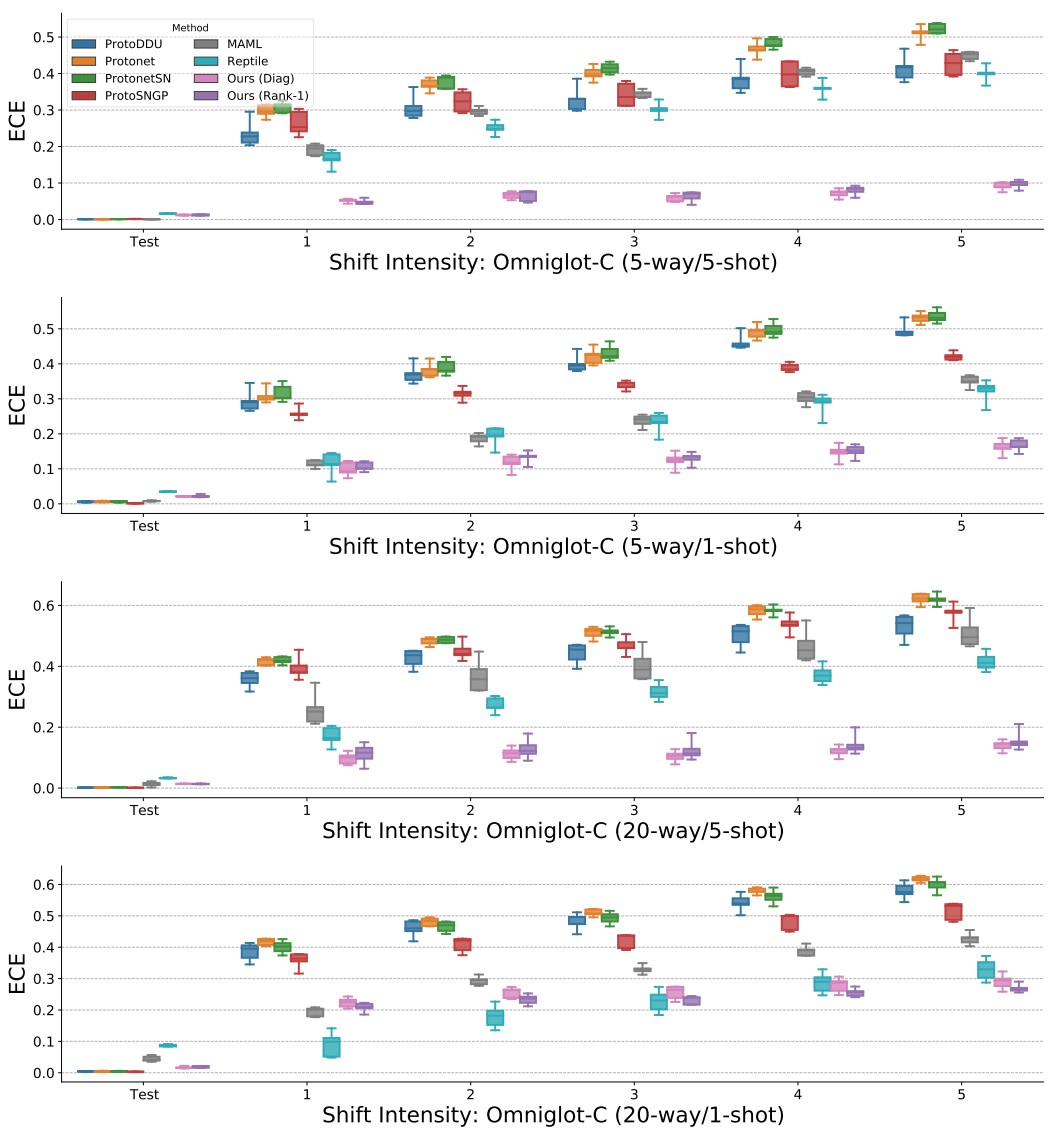

Figure 4: ECE results for all models on different variants of the Omniglot dataset. ProtoMahalanobis models show comparable in distribution ECE while significantly improving ECE over the baselines on corrupted instances from the dataset.

empirical features? 3) How does the ID/OOD calibration and accuracy compare with other popular baseline models?

**Datasets.** For few shot learning, we evaluate our model on both the Omniglot (Lake et al., 2015) and MiniImageNet (Vinyals et al., 2017) datasets. We utilize corrupted versions (Omniglot-C and MiniImageNet-C) which consists of 17 corruptions at 5 different intensities (Hendrycks & Dietterich, 2019). We follow the precedent set by Snell et al. (2017) and test Omniglot for 1000 random episodes and MiniImageNet for 600 episodes. For corruption experiments, the support set is uncorrupted, and corruption levels 0-5 are used as the query set (0 being the uncorrupted query set). We also experiment with multiple toy datasets which include half-moons, and concentric circles for binary classification and random 2D multivariate Gaussian distributions for multiclass classification (Figure 1). On the toy datasets, we create task distributions by sampling random tasks with biased support sets, applying random class shuffling and varying levels of noise added to each task. Randomly biasing each task ensures that no single task contains information from the whole distribution and therefore, the true distribution must be meta-learned through the episodic training over many such tasks. For a detailed explanation of the exact toy dataset task creation procedure, see the appendix section A.2.

**Baselines.** We compare our model against Protonets (Snell et al., 2017), A spectral normalized version of Protonets (Protonet-SN), MAML (Finn et al., 2017), Reptile (Nichol et al., 2018), and

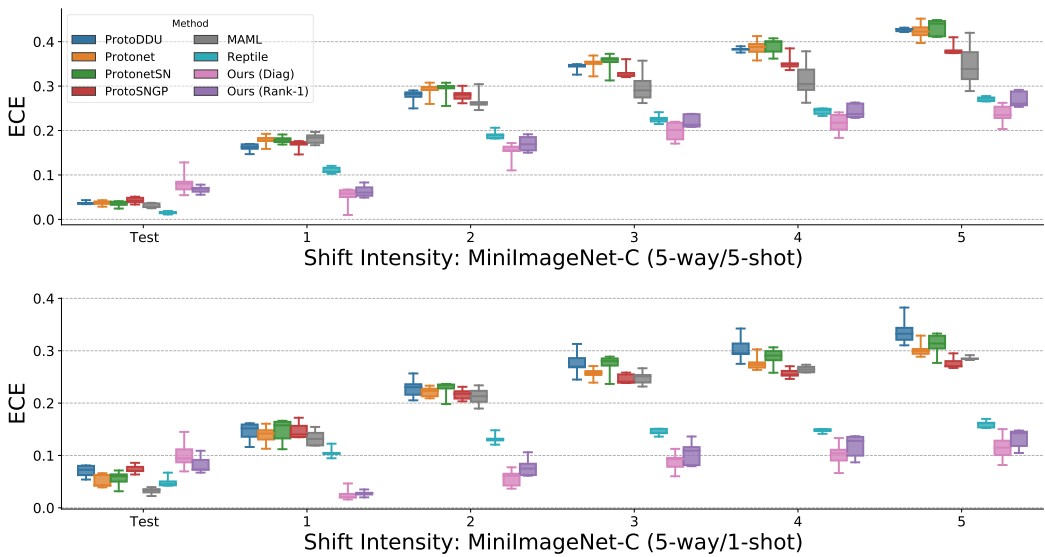

Figure 5: ECE for different variants of the MiniImageNet dataset. ProtoMahalanobis models show improved ECE on corrupted data instances while maintaining comparable performance on in-distribution data.

straightforward few-shot/protonet adaptations of Spectral Normalized Neural Gaussian Processes (ProtoSNGP) (Liu et al., 2020a) and Deep Deterministic Uncertainty (ProtoDDU) (Mukhoti et al., 2021). These models represent a range of both metric based, gradient based, and covariance based meta learning algorithms. All baseline models are temperature scaled after training, with the temperature parameter optimized via LBFGS for 50 iterations with a learning rate of $0.001$. This follows the temperature scaling implementation from Guo et al. (2017).

**Calibration Error.** We provide results for Expected Calibration Error (ECE) (Guo et al., 2017) on various types of OOD data in Figures 4 and 5 as well as Table 1. Accuracy and NLL are reported in Appendix A.8. Meta learning generally presents a high correlation between tasks, but random classes from different tasks which are not in the current support set $\mathcal{S}$ should still be treated as OOD. In Table 1 we provide results where the query set $\mathcal{Q}$ consists of random classes not in $\mathcal{S}$. ProtoMahalanobis models perform the best in every case except for Omniglot 20-way/1-shot, where Reptile showed the lowest ECE. The reason for this can be seen in Figure 4, where Reptile shows poor ID performance relative to all other models. Under-confidence on ID data can lead to better confidence scores on OOD data, even though the model is poorly calibrated. Likewise we also evaluate our models on Omniglot-C and MiniImageNet-C in Figures 4 and 5. As the corruption intensity increases, ProtoMahalanobis models exhibit lower ECE in relation to baseline models while maintaining competitive ID performance. Overall, Reptile shows the strongest calibration of baseline models although it can be underconfident on ID data as can be seen in Figure 4.

In our experiments, transductive batch normalization used in MAML/Reptile led to suboptimal results, as the normalization statistics depend on the query set which is simultaneously passed through the network. Passing a large batch of corrupted/uncorrupted samples caused performance degradation on ID data and presented an unrealistic setting. We therefore utilized the normalization scheme proposed by Nichol et al. (2018) which creates batch normalization statistics based on the whole support set plus a single query instance.

**Eigenvalue Distribution.** In Figure 6, we evaluate the effectiveness of meta learning the low rank covariance factors with $g_\phi$ by analyzing the eigenvalue distribution of both empirical covariance from DDU/SNGP and the encoded covariance from $g_\phi$ (Equation 5). The empirically calculated covariances exhibit lower diversity in eigenvalues, which implies that the learned Gaussian distribution is more spherical and uniform for every class. ProtoMahalanobis models, on the other hand, exhibit a more diverse range of eigenvalues, leading to non-trivial ellipsoid distributions. We also note that in addition to more diverse range of eigenvalues, the differences between the distributions of each class in $\mathcal{S}$ are also amplified in ProtoMahalanobis models, indicating a class specific variation between

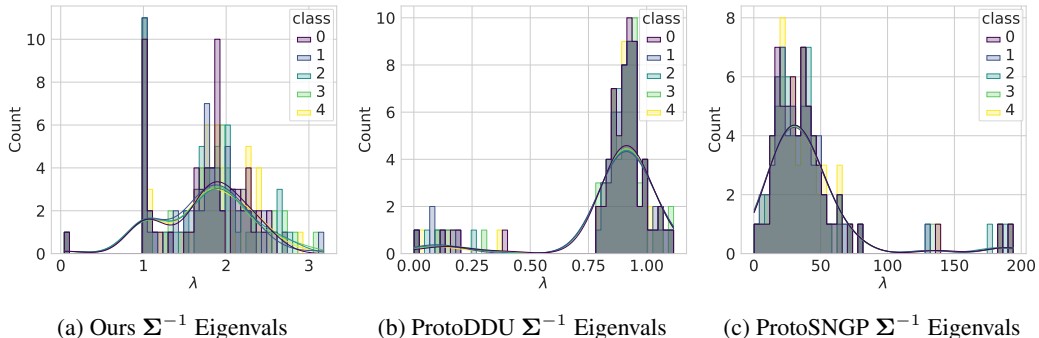

(a) Ours $\Sigma^{-1}$ Eigenvals     (b) ProtoDDU $\Sigma^{-1}$ Eigenvals     (c) ProtoSNGP $\Sigma^{-1}$ Eigenvals

Figure 6: Precision matrix eigenvalue distribution for various meta learning model variants. A diverse distribution of eigenvalues which varies by class, indicates a class specific, non-spherical Gaussian distribution is learned. Data comes from Omniglot 5-way/5-shot experiments.

learned covariance factors. Extra figures are reported in the Appendix A.7, where it can be seen that the eigenvalue distribution becomes less diverse for ProtoMahalanobis models in the one-shot setting.

**Architectures.** For both Omniglot and MiniImageNet experiments, we utilize a 4 layer convolutional neural network with 64 filters, followed by BatchNorm and ReLU nonlinearities. Each of the four layers is followed by a max-pooling layer which results in a vector embedding of size 64 for Omniglot and 1600 for MiniImageNet. Exact architectures can be found in Appendix A.9. Protonet-like models use BatchNorm with statistics tracked over the training set, and MAML-like baselines use Reptile Norm (Nichol et al., 2018). As spectral normalized models require residual connections to maintain the lower Lipschitz bound in equation 13, we add residual connections to the CNN architecture in all Protonet based models.

### 4.1 IMPLEMENTATION DETAILS

**ProtoSNGP & ProtoDDU** Both ProtoSNGP and ProtoDDU baselines are adapted to meta learning by using the original backbone implementation plus the addition of a positive constrained meta parameter for the first diagonal term in Equation 8 which is shared among all classes. This provides meta knowledge and a necessary first step in applying the recursive formula for inversion to make predictions on each query set seen during during training.

**Covariance Encoder $g_\phi$.** We utilize the Set Transformer (Lee et al., 2019), as the self-attention performed by the transformer is an expressive means to encode pairwise information between inputs. We initialize the seeds in the pooling layers (PMA), with samples from $\mathcal{N}(0, 1)$. We do not use any spectral normalization in $g_\phi$, as it should be sufficient to only require that the input to the encoder is composed of geometry preserving features. Crucially, we remove the residual connection $Q + \sigma(QK^\top)V$ as we found that this led to the pooling layer ignoring the inputs and outputting an identical covariance for each class in each task. In the one-shot case, we skip the centering about the centroid in Equation 4 because it would place all class centroids at the origin.

### 5 CONCLUSION

It is widely known that DNNs can be miscalibrated for OOD data. We have shown that existing covariance based uncertainty quantification methods fail to calibrate well when given a limited amounts of data for class-specific covariance construction for meta learning. In this work, we have proposed a novel method which meta-learns a diagonal or diagonal plus low rank covariance matrix which can be used for downstream tasks such as uncertainty calibration. Additionally, we have proposed an inference procedure and energy tuning scheme which can overcome miscalibration due to the shift invariance property of softmax. We further enforce bi-Lipschitz regularization of neural network layers to preserve relative distances between data instances in the feature spaces. We validated our methods on both synthetic data and two benchmark few-shot learning datasets, showing that the final predictive distribution of our method is well calibrated under a distributional dataset shift when compared with relevant baselines.

## 6 ACKNOWLEDGEMENTS

This work was supported by the Institute of Information & communications Technology Planning & Evaluation (IITP) grant funded by the Korea government(MSIT) (No.2019-0-00075, Artificial Intelligence Graduate School Program(KAIST)), the Engineering Research Center Program through the National Research Foundation of Korea (NRF) funded by the Korean Government MSIT (NRF-2018R1A5A1059921), the Institute of Information & communications Technology Planning & Evaluation (IITP) grant funded by the Korea government (MSIT) No. 2021-0-02068 (Artificial Intelligence Innovation Hub), and the National Research Foundation of Korea (NRF) funded by the Ministry of Education (NRF2021R1F1A1061655).

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

## A  APPENDIX

### A.1  LOSS DERIVATION (EQUATION 7)

The full derivation of Equation 7 can be achieved by first applying Bayes' Rule, assuming a simple uniform prior over the class labels, $p(y_i|\mathbf{x}_i)$ can be proportionately expressed as,

$$p(y_i|\mathbf{x}_i) = \frac{p(\mathbf{x}_i|y_k)p(y_k)}{p(\mathbf{x}_i)} \propto p(\mathbf{x}_i|y_k)p(y_k) \tag{14}$$

In which case the objective of the model becomes raising the class conditional $p(\mathbf{x}_i|y_i)$, while simultaneously lowering $p(\mathbf{x}_i|y_j) \ \forall \ j \neq i$. This is in fact equivalent to a softmax + cross entropy loss over the class conditional densities which are output from our model. In the softmax case, maximizing $p(y_i|\mathbf{x}_i)$ for a given class can be done by,

$$p(y_i|\mathbf{x}_i) = \frac{e^{z_i}}{\sum_{z'} e^{z'}} \tag{15}$$

Which them implies that the loss to be minimized is the following, commonly known as the negative log likelihood of the data, or the empirical cross entropy between the true data distribution and the predictive distribution of the model.

$$
\begin{aligned}
\mathcal{L}_{NLL} &= \mathbb{E}_{\mathcal{D}}[-\log p(y|\mathbf{x})] \\
&= \frac{1}{N}\sum_{i=0}^{N} -\log p(y_i|\mathbf{x}_i) \\
&= \frac{1}{N}\sum_{i=0}^{N} \left( -z_j + \log\sum_{z'_j}\exp(z'_j) \right)_i \\
\mathcal{L}_{CE} &= -\int_{\mathbf{x}} p_{\mathbf{x}}(\mathbf{x})\log p_\theta(y|\mathbf{x})d\mathbf{x} \\
&\approx \frac{1}{N}\sum_{i=0}^{N} -\log p_\theta(y_i|\mathbf{x}_i) \\
&= \frac{1}{N}\sum_{i=0}^{N} \left( -z_j + \log\sum_{z'_j}\exp(z'_j) \right)_i
\end{aligned}
\tag{16}
$$

In our case, assuming a uniform prior over the classes, we can analogously formulate the loss as,

$$
\begin{aligned}
\mathcal{L} &= \mathbb{E}_{\mathcal{D}}[-\log p(\mathbf{x}|y)p(y)] \\
&= \frac{1}{N}\sum_{i=0}^{N} \left( -\log p(\mathbf{x}_i|y_k)p(y_k) + \log\sum_{\mathbf{x}'} p(\mathbf{x}'_i|y_k)p(y_k) \right) \\
&= \frac{1}{N}\sum_{i=0}^{N} \left( -\log p(\mathbf{x}_i|y_k) - \log p(y_k) + \log p(y_k) + \log\sum_{\mathbf{x}'} p(\mathbf{x}'_i|y_k) \right) \\
&= \frac{1}{N}\sum_{i=0}^{N} \left( -\log p(\mathbf{x}_i|y_k) + \log\sum_{\mathbf{x}'} p(\mathbf{x}'_i|y_k) \right)
\end{aligned}
\tag{17}
$$

### A.2  TOY DATASETS

To add bias to each samples task from our 2D toy datasets, we first randomly choose an axis (X or Y) for each class and then slice the datapoints in half randomly. We then sample the support set from the

chosen biased subset and leave the rest of the remaining points for the query set. Each sampled task calculates the mean and variance from the support set, which are then used to normalize all instances in $\mathcal{S}$ and $\mathcal{Q}$.

| Dataset | N-Way | K-Shot |
|---------|-------|--------|
| Circles | 2 | 5 |
| Moons | 2 | 5 |
| Gaussians | 10 | 10 |

### A.2.1 META MOONS

For the Meta Moons dataset, we randomly invert the classes to make sure that the class indices appear in a random order for each task. We add a random amount of Gaussian noise to each moon with a uniform standard deviation in the range of $(0, 0.25]$.

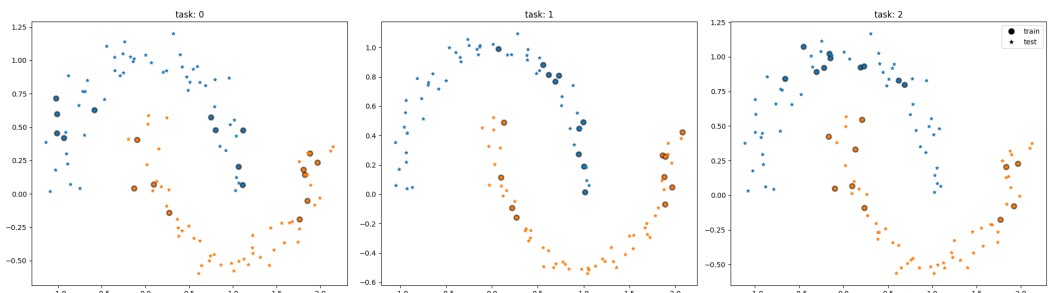

Figure 7: Random task samples from the Meta Moons dataset.

### A.2.2 META CIRCLES

For the Meta Circles dataset, we randomly invert the order of the classes so that the inner circle and the outer circle are not guaranteed to appear in the same order on every task. We inject a random amount of Gaussian noise into the data, with a uniformly random standard deviation in the range of $(0, 0.25]$. We also randomly choose the scale factor between the size of the inner circle and the outer circle, which is uniformly random in the range of $(0, 0.8]$

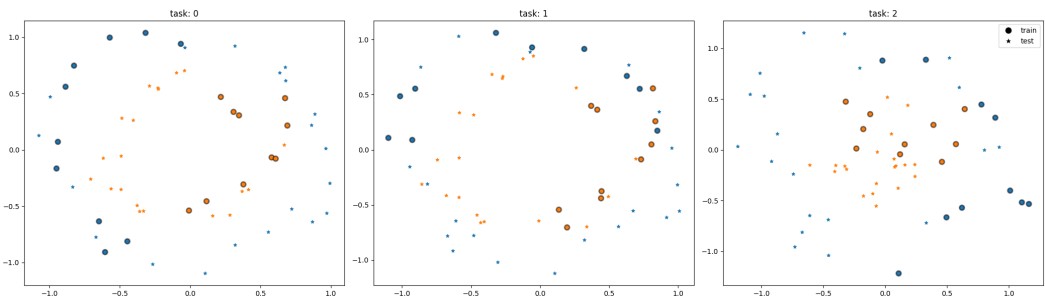

Figure 8: Random task samples from the Meta Circles dataset.

### A.2.3 META GAUSSIANS

The task construction of the Meta Gaussians dataset requires that we construct random positive semidefinite covariance matrices for each class. We first uniformly sample $N$ $2 \times 2$ matrices in the range $U(-1, 1)$ and perform a QR decomposition to extract orthonormal matrices $Q$. We then sample a random diagonal $D \sim U(0, 1)$, and construct the final matrix as $QDQ^\top$ which is positive

Table 2: Accuracy, ECE, NLL, and OOD AUPR for different n-way k-shot classification problems on the Omniglot-C dataset which contains 17 different corruptions at 5 different intensity levels. MAML/Reptile both utilize 'Reptile Norm' instead of transductive BatchNorm

| | Accuracy ↑ | | | | NLL ↓ | | | |
|---|---|---|---|---|---|---|---|---|
| Model | 5-way 5-shot | 5-way 1-shot | 20-way 5-shot | 20-way 1-shot | 5-way 5-shot | 5-way 1-shot | 20-way 5-shot | 20-way 1-shot |
| MAML | 65.02±17.94 | 64.72±16.96 | 48.17±23.93 | 44.35±22.27 | 3.658±2.421 | 1.528±0.869 | 5.962±3.822 | 3.668±1.873 |
| Reptile | 61.29±18.61 | 60.01±17.63 | 46.55±23.75 | 43.42±22.13 | 2.300±1.345 | 1.571±0.796 | 3.937±2.357 | 2.864±1.380 |
| Protonet | 60.91±19.13 | 58.15±19.68 | 46.29±25.21 | 43.11±25.55 | 6.526±3.800 | 6.539±4.029 | 10.706±5.689 | 9.119±4.890 |
| Protonet-SN | 60.08±19.47 | 57.64±19.89 | 46.19±25.22 | 43.47±25.44 | 7.189±4.261 | 6.541±3.960 | 11.200±6.168 | 8.446±4.590 |
| ProtoDDU | 60.31±19.19 | 58.03±19.57 | 45.75±25.33 | 43.90±25.22 | 10.945±7.143 | 10.428±7.186 | 18.014±9.740 | 17.039±9.935 |
| ProtoSNGP | 59.18±19.66 | 57.03±19.91 | 46.49±25.12 | 44.37±24.85 | 2.534±1.302 | 2.015±0.967 | 6.409±3.196 | 4.151±1.982 |
| Ours (Diag) | 60.77±19.12 | 57.71±19.88 | 45.98±25.31 | 43.05±25.59 | 1.010±0.476 | 1.205±0.547 | 2.466±1.141 | 3.854±1.775 |
| Ours (Rank 1) | 59.88±19.55 | 58.15±19.59 | 45.47±25.57 | 43.12±25.57 | 1.020±0.481 | 1.312±0.595 | 2.541±1.212 | 3.564±1.624 |
| Ours (Rank 2) | 59.68±19.61 | 56.61±20.21 | 45.53±25.52 | 43.28±25.45 | 1.045±0.495 | 1.314±0.594 | 2.561±1.194 | 3.787±1.760 |
| Ours (Rank 4) | 60.28±19.51 | 59.14±19.32 | 45.87±25.44 | 42.38±25.90 | 1.068±0.510 | 1.263±0.580 | 2.528±1.168 | 4.019±1.877 |
| Ours (Rank 8) | 59.69±19.69 | 58.21±19.63 | 45.93±25.39 | 43.53±25.42 | 1.055±0.501 | 1.329±0.613 | 2.486±1.156 | 3.474±1.607 |
| | ECE ↓ | | | | OOD AUPR | | | |
| MAML | 28.10±15.25 | 19.96±11.74 | 33.64±17.35 | 27.95±13.12 | 0.602±0.060 | 0.653±0.075 | 0.440±0.032 | 0.484±0.098 |
| Reptile | 24.85±13.12 | 19.69±10.29 | 26.33±13.33 | 19.98±9.77 | 0.617±0.081 | 0.672±0.098 | 0.646±0.087 | 0.720±0.113 |
| Protonet | 34.08±17.02 | 35.69±17.62 | 43.57±20.89 | 43.47±20.69 | 0.867±0.169 | 0.853±0.163 | 0.876±0.172 | 0.863±0.166 |
| Protonet-SN | 35.12±17.46 | 36.27±17.85 | 43.75±20.89 | 42.06±20.10 | 0.869±0.169 | 0.857±0.164 | 0.875±0.171 | 0.862±0.166 |
| ProtoDDU | 27.84±14.19 | 33.79±16.61 | 37.62±18.16 | 40.83±19.55 | 0.675±0.083 | 0.552±0.044 | 0.647±0.074 | 0.579±0.062 |
| ProtoSNGP | 29.26±14.53 | 28.70±14.07 | 40.42±19.46 | 36.43±17.36 | 0.875±0.172 | 0.855±0.163 | 0.878±0.173 | 0.858±0.164 |
| Ours (Diag) | 5.87±2.58 | 11.15±4.93 | 9.81±4.36 | 21.99±9.65 | 0.869±0.169 | 0.855±0.164 | 0.876±0.172 | 0.863±0.166 |
| Ours (Rank 1) | 6.10±2.87 | 11.81±4.96 | 11.30±5.48 | 20.26±8.68 | 0.869±0.169 | 0.850±0.162 | 0.874±0.171 | 0.863±0.166 |
| Ours (Rank 2) | 6.62±3.09 | 12.38±5.11 | 11.49±5.12 | 21.42±9.45 | 0.867±0.169 | 0.853±0.163 | 0.874±0.171 | 0.862±0.166 |
| Ours (Rank 4) | 7.10±3.40 | 11.17±4.95 | 10.87±4.75 | 22.90±10.14 | 0.868±0.169 | 0.851±0.162 | 0.874±0.171 | 0.864±0.167 |
| Ours (Rank 8) | 7.05±3.42 | 12.89±5.60 | 10.61±4.84 | 20.56±9.22 | 0.868±0.169 | 0.851±0.162 | 0.876±0.172 | 0.863±0.166 |

semi-definite. This leads to the distribution of each class being an elliptical multivariate Gaussian distribution.

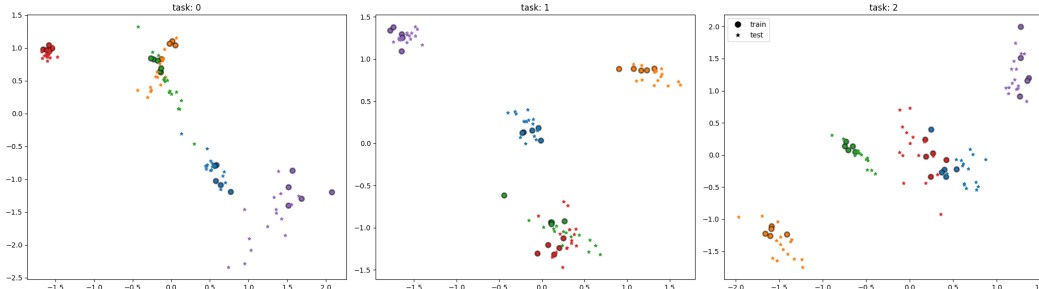

Figure 9: Random task samples from the Meta Gaussians dataset.

## A.3 EXTRA RESULTS

We provide extra results on the MiniImageNet-C and Omniglot-C dataset here. Tables 2 and 3 contain results averaged over the whole corrupted dataset, including the natural test set and all 5 levels of corruption

## A.4 SET ENCODING RELATED WORKS

Set encoding functions require special end-to-end design considerations such as obeying permutation invariance w.r.t. the input set $f(\{\mathbf{X}_1, \mathbf{X}_2, ..., \mathbf{X}_n\}) = f(\{\mathbf{X}_{\pi(1)}, \mathbf{X}_{\pi(2)}, ..., \mathbf{X}_{\pi(n)}\})$ for any random permutation of indices $\pi(.)$. Likewise, the intermediate latent representations must satisfy permutation equivariance such that $f(\{\mathbf{X}_{\pi(1)}, \mathbf{X}_{\pi(2)}, ..., \mathbf{X}_{\pi(n)}\}) = \{f_{\pi(1)}(\mathbf{X}), f_{\pi(2)}(\mathbf{X}), ..., f_{\pi(1)}(\mathbf{X})\}$.

Deepsets (Zaheer et al., 2017) first proposed basic adaptations of linear and convolutional neural networks which obey the above required properties and have the addition of a sum decomposable (permutation invariant) pooling function and decoder to match the requirements of the given task. As sets can have complex interactions between elements, it may be beneficial to model pairwise interactions between set elements. The Set Transformer (Lee et al., 2019) uses a transformer architecture with self attention to model such pairwise interactions between set elements. As

Table 3: Accuracy, ECE, NLL, and AUPR for different n-way k-shot classification problems on the MiniImageNet-C dataset which contains 17 different corruptions at 5 different intensity levels. MAML/Reptile both utilize 'Reptile Norm' instead of transductive BatchNorm

| | Accuracy ↑ | | NLL ↓ | |
|---|---|---|---|---|
| Model | 5-way 1-shot | 5-way 5-shot | 5-way 1-shot | 5-way 5-shot |
| MAML | 31.94±7.85 | 39.30±12.92 | 1.720±0.242 | 1.748±0.462 |
| Reptile | 33.07±7.85 | 39.18±12.84 | 1.580±0.150 | 1.581±0.339 |
| Protonet | 33.43±8.49 | 41.35±14.03 | 1.801±0.323 | 2.123±0.798 |
| Protonet-SN | 32.79±8.56 | 40.95±14.21 | 1.836±0.341 | 2.112±0.802 |
| ProtoDDU | 33.62±8.92 | 41.46±14.35 | 1.906±0.465 | 2.180±0.932 |
| ProtoSNGP | 33.56±8.65 | 41.20±13.66 | 1.699±0.267 | 1.889±0.614 |
| Ours (Diag) | 33.21±8.68 | 40.69±13.66 | 1.556±0.158 | 1.630±0.423 |
| Ours (Rank 1) | 33.19±8.45 | 40.89±13.90 | 1.575±0.171 | 1.699±0.484 |
| Ours (Rank 2) | 33.03±8.54 | 40.90±13.87 | 1.571±0.174 | 1.659±0.456 |
| Ours (Rank 4) | 32.52±8.45 | 41.24±13.84 | 1.591±0.191 | 1.696±0.486 |
| Ours (Rank 8) | 32.41±8.55 | 40.33±13.77 | 1.581±0.175 | 1.644±0.442 |
| | ECE ↓ | | AUPR ↑ | |
| MAML | 19.62±9.02 | 24.03±11.29 | 0.536±0.083 | 0.628±0.106 |
| Reptile | 12.32±3.84 | 17.60±8.98 | 0.756±0.131 | 0.749±0.124 |
| Protonet | 20.78±8.98 | 27.75±13.62 | 0.637±0.094 | 0.579±0.071 |
| Protonet-SN | 21.66±9.28 | 27.99±13.90 | 0.629±0.085 | 0.572±0.068 |
| ProtoDDU | 22.70±9.69 | 27.13±13.71 | 0.530±0.041 | 0.620±0.061 |
| ProtoSNGP | 20.30±7.28 | 25.96±12.19 | 0.636±0.080 | 0.653±0.089 |
| Ours (Diag) | 8.13±3.72 | 15.57±7.32 | 0.636±0.092 | 0.578±0.065 |
| Ours (Rank 1) | 9.02±3.83 | 17.27±8.42 | 0.625±0.087 | 0.574±0.067 |
| Ours (Rank 2) | 9.27±4.02 | 16.27±7.33 | 0.632±0.087 | 0.581±0.064 |
| Ours (Rank 4) | 9.87±5.29 | 16.71±7.85 | 0.629±0.085 | 0.569±0.058 |
| Ours (Rank 8) | 9.27±4.38 | 15.93±7.50 | 0.637±0.092 | 0.574±0.069 |

Table 4: Accuracy, ECE, NLL, and AUPR for different n-way k-shot classification problems on the Omniglot dataset. All metrics are measured on the natural test set except AUPR/AUROC which is measured using random classes which are different from the classes in the support set. Our model maintains competitive performance for ID data on all metrics. MAML/Reptile both utilize 'Reptile Norm' instead of transductive BatchNorm

| | Accuracy ↑ | | | | NLL ↓ | | | |
|---|---|---|---|---|---|---|---|---|
| Model | 5-way 5-shot | 5-way 1-shot | 20-way 5-shot | 20-way 1-shot | 5-way 5-shot | 5-way 1-shot | 20-way 5-shot | 20-way 1-shot |
| MAML | 99.51±0.06 | 96.55±0.23 | 97.96±0.28 | 91.97±0.27 | 0.015±0.002 | 0.104±0.006 | 0.078±0.015 | 0.289±0.013 |
| Reptile | 98.55±0.07 | 95.72±0.38 | 96.50±0.07 | 90.95±0.47 | 0.054±0.002 | 0.150±0.010 | 0.142±0.002 | 0.365±0.015 |
| Protonet | 99.65±0.02 | 98.24±0.16 | 99.29±0.05 | 97.47±0.09 | 0.013±0.002 | 0.059±0.007 | 0.027±0.006 | 0.087±0.008 |
| Protonet-SN | 99.67±0.04 | 98.26±0.12 | 99.26±0.06 | 97.51±0.16 | 0.013±0.003 | 0.061±0.007 | 0.029±0.006 | 0.086±0.011 |
| ProtoDDU | 99.70±0.05 | 98.37±0.11 | 99.28±0.05 | 97.54±0.16 | 0.010±0.002 | 0.058±0.010 | 0.027±0.005 | 0.085±0.010 |
| ProtoSNGP | 99.65±0.07 | 98.23±0.08 | 99.23±0.06 | 97.41±0.13 | 0.012±0.003 | 0.054±0.004 | 0.029±0.006 | 0.085±0.006 |
| Ours (Diag) | 99.64±0.06 | 98.21±0.23 | 99.26±0.01 | 97.49±0.09 | 0.020±0.002 | 0.064±0.005 | 0.032±0.002 | 0.089±0.006 |
| Ours (Rank 1) | 99.63±0.06 | 98.21±0.12 | 99.29±0.03 | 97.61±0.14 | 0.020±0.002 | 0.067±0.005 | 0.031±0.002 | 0.086±0.007 |
| Ours (Rank 2) | 99.62±0.06 | 98.30±0.23 | 99.30±0.06 | 97.56±0.10 | 0.020±0.002 | 0.064±0.010 | 0.031±0.003 | 0.087±0.007 |
| Ours (Rank 4) | 99.66±0.04 | 98.42±0.17 | 99.28±0.07 | 97.56±0.17 | 0.019±0.002 | 0.060±0.005 | 0.032±0.003 | 0.088±0.007 |
| Ours (Rank 8) | 99.64±0.02 | 98.35±0.16 | 99.32±0.04 | 97.63±0.16 | 0.019±0.001 | 0.059±0.004 | 0.030±0.003 | 0.084±0.008 |
| | ECE ↓ | | | | OOD AUPR ↑ | | | |
| MAML | 0.05±0.03 | 1.06±0.12 | 1.39±0.87 | 4.95±0.68 | 0.856±0.006 | 0.799±0.010 | 0.622±0.023 | 0.578±0.007 |
| Reptile | 1.64±0.08 | 3.95±0.41 | 3.21±0.11 | 8.73±0.12 | 0.831±0.019 | 0.813±0.015 | 0.591±0.002 | 0.579±0.003 |
| Protonet | 0.09±0.02 | 0.54±0.09 | 0.19±0.04 | 0.35±0.09 | 0.994±0.001 | 0.977±0.001 | 0.990±0.001 | 0.974±0.001 |
| Protonet-SN | 0.09±0.04 | 0.51±0.17 | 0.21±0.03 | 0.39±0.11 | 0.994±0.000 | 0.977±0.002 | 0.990±0.000 | 0.975±0.002 |
| ProtoDDU | 0.07±0.02 | 0.42±0.15 | 0.14±0.03 | 0.33±0.13 | 0.482±0.003 | 0.475±0.004 | 0.496±0.002 | 0.496±0.002 |
| ProtoSNGP | 0.09±0.04 | 0.15±0.04 | 0.14±0.05 | 0.20±0.04 | 0.994±0.001 | 0.977±0.003 | 0.989±0.001 | 0.972±0.002 |
| Ours (Diag) | 1.07±0.12 | 2.02±0.12 | 1.13±0.05 | 1.58±0.26 | 0.994±0.001 | 0.976±0.003 | 0.990±0.000 | 0.974±0.001 |
| Ours (Rank 1) | 1.06±0.09 | 2.14±0.30 | 1.14±0.05 | 1.71±0.27 | 0.994±0.001 | 0.976±0.002 | 0.990±0.001 | 0.974±0.001 |
| Ours (Rank 2) | 1.07±0.10 | 2.11±0.38 | 1.13±0.08 | 1.47±0.28 | 0.994±0.001 | 0.977±0.004 | 0.990±0.001 | 0.975±0.002 |
| Ours (Rank 4) | 1.00±0.11 | 2.00±0.45 | 1.16±0.06 | 1.52±0.12 | 0.994±0.000 | 0.977±0.002 | 0.990±0.001 | 0.975±0.001 |
| Ours (Rank 8) | 1.03±0.08 | 1.88±0.17 | 1.10±0.10 | 1.47±0.11 | 0.994±0.000 | 0.978±0.002 | 0.990±0.001 | 0.974±0.001 |

Table 5: Accuracy, NLL, ECE, and AUPR for different n-way k-shot classification problems on the Mini-ImageNet dataset. All metrics are measured on the natural test set except AUPR/AUROC which is measured using random classes which are different from the classes in the support set. Our model maintains competitive performance for ID data on all metrics. MAML/Reptile both utilize 'Reptile Norm' instead of transductive BatchNorm

| | Accuracy ↑ | | NLL ↓ | |
|---|---|---|---|---|
| Model | 5-way 1-shot | 5-way 5-shot | 5-way 1-shot | 5-way 5-shot |
| MAML | 46.13±1.19 | 64.71±0.50 | 1.297±0.015 | 0.921±0.017 |
| Reptile | 47.79±1.21 | 62.89±0.88 | 1.297±0.023 | 0.967±0.017 |
| Protonet | 48.61±0.91 | 67.57±0.55 | 1.245±0.019 | 0.832±0.008 |
| Protonet-SN | 47.47±0.90 | 68.03±0.79 | 1.279±0.017 | 0.820±0.016 |
| ProtoDDU | 49.57±0.53 | 68.31±0.59 | 1.246±0.006 | 0.816±0.016 |
| ProtoSNGP | 49.55±0.90 | 66.89±0.88 | 1.232±0.012 | 0.841±0.020 |
| Ours (Diag) | 48.31±0.39 | 66.12±1.76 | 1.277±0.023 | 0.887±0.044 |
| Ours (Rank 1) | 48.57±0.96 | 66.54±0.66 | 1.267±0.013 | 0.859±0.016 |
| Ours (Rank 2) | 48.08±0.99 | 67.17±0.56 | 1.271±0.021 | 0.853±0.017 |
| Ours (Rank 4) | 47.76±0.62 | 66.73±0.37 | 1.274±0.020 | 0.868±0.010 |
| Ours (Rank 8) | 48.91±0.87 | 66.58±1.67 | 1.272±0.039 | 0.873±0.038 |

| | ECE ↓ | | AUPR ↑ | |
|---|---|---|---|---|
| MAML | 3.39±1.09 | 2.79±0.45 | 0.508±0.009 | 0.544±0.005 |
| Reptile | 4.65±0.76 | 1.64±0.23 | 0.517±0.006 | 0.542±0.003 |
| Protonet | 5.62±1.08 | 4.09±0.86 | 0.609±0.010 | 0.596±0.006 |
| Protonet-SN | 6.77±1.81 | 3.22±0.96 | 0.608±0.011 | 0.602±0.002 |
| ProtoDDU | 7.33±1.44 | 3.46±0.55 | 0.473±0.006 | 0.479±0.003 |
| ProtoSNGP | 7.21±1.26 | 4.06±0.84 | 0.621±0.007 | 0.687±0.004 |
| Ours (Diag) | 9.40±1.82 | 8.32±2.46 | 0.607±0.011 | 0.602±0.013 |
| Ours (Rank 1) | 8.09±1.60 | 6.77±0.53 | 0.605±0.007 | 0.611±0.008 |
| Ours (Rank 2) | 7.50±2.30 | 7.92±1.34 | 0.610±0.008 | 0.611±0.003 |
| Ours (Rank 4) | 7.57±2.45 | 8.02±2.19 | 0.603±0.005 | 0.609±0.005 |
| Ours (Rank 8) | 10.03±3.64 | 8.80±1.53 | 0.609±0.010 | 0.606±0.005 |

transformers have a quadratic complexity w.r.t. input set length, it may not be possible to process a large set with a transformer and maintain permutation invariance, if the set will not fit into memory. Therefore, recent works have also further explored how to make an attentive set encoder which can process sets in batches (Andreis et al., 2021) while maintaining the above requirements of set functions.

For our model, we chose to use the set transformer architecture, as it models pairwise interactions between elements which is an implicit requirement of construction a Gaussian covariance matrix. Therefore, it has the proper inductive biases needed to satisfy our requirement of predicting low rank covariance factors given an input set of features.

## A.5 EXTRA TOY RESULTS

In Figures 10, 11, 12, 13, 14, 15, and 16 we provide extra qualitative results on toy dataset covariances and entropy surfaces. In Tables 6, 7, and 8 we provide tabular results of all toy experiments, showcasing the differences between in-distribution data and random uniform OOD noise.

| | In Distribiution | | | Out of Distribiution | | |
|---|---|---|---|---|---|---|
| Model | Accuracy ↑ | NLL ↓ | ECE ↓ | ECE ↓ | AUPR ↑ | AUROC ↑ |
| Protonet | 97.02±1.60 | 0.171±0.110 | **2.21±1.40** | 48.16±0.91 | 0.999±0.000 | 0.931±0.012 |
| ProtonetSN | **97.31±1.54** | 0.140±0.088 | 2.38±1.38 | 48.40±0.64 | 0.999±0.000 | 0.932±0.006 |
| Proto DDU | 96.04±2.74 | 0.158±0.077 | 2.43±1.04 | 49.43±0.37 | 0.976±0.001 | 0.119±0.013 |
| Proto SNGP | 97.22±1.19 | **0.138±0.046** | 4.81±2.63 | 45.74±4.19 | 0.995±0.001 | 0.684±0.064 |
| Ours (Diag) | 96.82±1.09 | 0.167±0.056 | 5.09±1.18 | **15.66±2.65** | 0.999±0.000 | 0.937±0.006 |
| Ours (Rank-1) | 96.86±1.42 | 0.157±0.049 | 4.21±1.77 | 20.60±2.24 | 0.999±0.000 | 0.934±0.008 |
| Ours (Rank-2) | 96.90±1.55 | 0.162±0.042 | 5.13±1.39 | 17.74±2.76 | 0.999±0.000 | 0.939±0.006 |
| Ours (Rank-4) | 96.90±1.15 | 0.157±0.033 | 4.69±1.32 | 18.49±2.62 | 0.999±0.000 | 0.937±0.006 |
| Ours (Rank-8) | 96.69±1.36 | 0.171±0.047 | 4.74±1.48 | 19.75±3.53 | 0.999±0.000 | 0.935±0.007 |
| Ours (Rank-16) | 96.73±1.28 | 0.161±0.057 | 3.80±1.87 | 18.47±2.30 | 0.999±0.000 | 0.939±0.007 |
| Ours (Rank-32) | 96.73±1.28 | 0.170±0.042 | 4.49±1.01 | 18.22±3.26 | 0.999±0.000 | **0.939±0.005** |
| Ours (Rank-64) | 96.73±1.55 | 0.159±0.045 | 4.43±0.99 | 17.90±2.49 | 0.999±0.000 | 0.938±0.005 |

Table 6: Tabular results from the meta-moons toy experiment

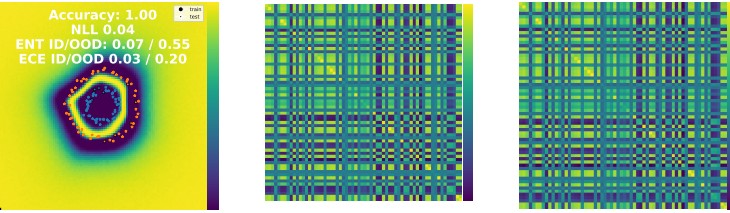

(a) Meta Circles (ProtoMahalanobisFC). From left to right: entropy surface, covariances for class 1-2

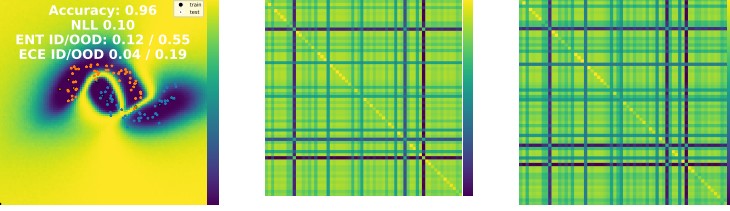

(b) Meta Gaussians (ProtoMahalanobisFC) From left to right: entropy surface, covariances for class 1-2

Figure 10: ProtoMahalanobisFC model performance on the meta-moons and meta-circles toy datasets.

| Model | In Distribiution | | | Out of Distribiution | | |
| --- | --- | --- | --- | --- | --- | --- |
| | Accuracy ↑ | NLL ↓ | ECE ↓ | ECE ↓ | AUPR ↑ | AUROC ↑ |
| Protonet | 89.12±3.92 | 0.343±0.091 | 4.33±0.83 | 81.84±0.78 | 0.999±0.000 | 0.950±0.009 |
| ProtonetSN | 88.93±4.12 | 0.342±0.098 | 4.23±0.71 | 82.42±0.87 | 0.999±0.000 | 0.951±0.008 |
| Proto DDU | 90.45±2.67 | 0.267±0.061 | **3.52±0.95** | 84.48±0.41 | 0.969±0.001 | 0.175±0.003 |
| Proto SNGP | 89.63±3.30 | 0.323±0.063 | 7.39±2.71 | 62.60±6.57 | 0.999±0.000 | 0.916±0.014 |
| Ours (Diag) | 90.61±3.54 | 0.279±0.075 | 7.24±1.57 | **42.55±0.98** | 0.999±0.000 | 0.954±0.005 |
| Ours (Rank-1) | 91.01±2.65 | 0.271±0.062 | 7.12±1.00 | 42.77±1.62 | 0.999±0.000 | 0.954±0.005 |
| Ours (Rank-2) | **91.31±2.87** | 0.269±0.064 | 7.59±1.34 | 43.13±1.90 | 0.999±0.000 | 0.955±0.004 |
| Ours (Rank-4) | 90.96±2.76 | 0.272±0.068 | 6.68±0.56 | 43.33±1.80 | 0.999±0.000 | 0.955±0.004 |
| Ours (Rank-8) | 90.99±2.52 | 0.268±0.065 | 6.74±1.02 | 43.14±1.49 | 0.999±0.000 | 0.955±0.004 |
| Ours (Rank-16) | 91.01±2.88 | 0.264±0.065 | 6.96±1.23 | 42.93±1.94 | 0.999±0.000 | 0.955±0.005 |
| Ours (Rank-32) | 90.45±3.22 | 0.267±0.069 | 6.35±0.27 | 42.89±1.48 | 0.999±0.000 | **0.956±0.004** |
| Ours (Rank-64) | 90.83±2.99 | **0.264±0.064** | 6.79±0.83 | 42.67±1.67 | 0.999±0.000 | **0.956±0.004** |

Table 7: Tabular results from the meta-Gaussians toy experiment

| Model | In Distribiution | | | Out of Distribiution | | |
| --- | --- | --- | --- | --- | --- | --- |
| | Accuracy ↑ | NLL ↓ | ECE ↓ | ECE ↓ | AUPR ↑ | AUROC ↑ |
| Protonet | 94.45±3.18 | 0.195±0.106 | 3.49±1.67 | 49.19±0.25 | 1.000±0.000 | 0.952±0.007 |
| ProtonetSN | 94.53±2.49 | 0.185±0.098 | **3.03±1.58** | 49.17±0.21 | 1.000±0.000 | 0.952±0.007 |
| Proto DDU | **95.02±1.79** | **0.165±0.088** | 3.62±2.27 | 48.87±0.18 | 0.972±0.001 | 0.072±0.008 |
| Proto SNGP | 94.49±2.09 | 0.192±0.071 | 6.05±3.56 | 45.14±3.08 | 0.992±0.001 | 0.683±0.055 |
| Ours (Diag) | 94.24±3.85 | 0.215±0.139 | 4.11±0.62 | **14.64±4.32** | 1.000±0.000 | 0.954±0.011 |
| Ours (Rank-1) | 94.08±4.62 | 0.214±0.158 | 4.34±1.46 | 19.04±8.21 | 1.000±0.000 | 0.953±0.013 |
| Ours (Rank-2) | 94.53±4.27 | 0.192±0.148 | 3.54±1.52 | 18.22±3.72 | 1.000±0.000 | 0.954±0.013 |
| Ours (Rank-4) | 94.12±4.68 | 0.209±0.158 | 4.27±1.83 | 19.61±4.52 | 1.000±0.000 | 0.954±0.013 |
| Ours (Rank-8) | 94.00±4.69 | 0.194±0.134 | 3.80±1.45 | 19.37±4.42 | 1.000±0.000 | 0.955±0.014 |
| Ours (Rank-16) | 94.12±4.40 | 0.205±0.148 | 4.12±1.93 | 20.59±5.47 | 1.000±0.000 | 0.955±0.013 |
| Ours (Rank-32) | 93.84±4.66 | 0.193±0.141 | 3.48±1.54 | 20.50±5.56 | 1.000±0.000 | 0.954±0.014 |
| Ours (Rank-64) | 94.16±4.61 | 0.196±0.146 | 3.46±1.31 | 19.53±6.30 | 1.000±0.000 | 0.955±0.014 |

Table 8: Tabular results from the meta-circles toy experiment

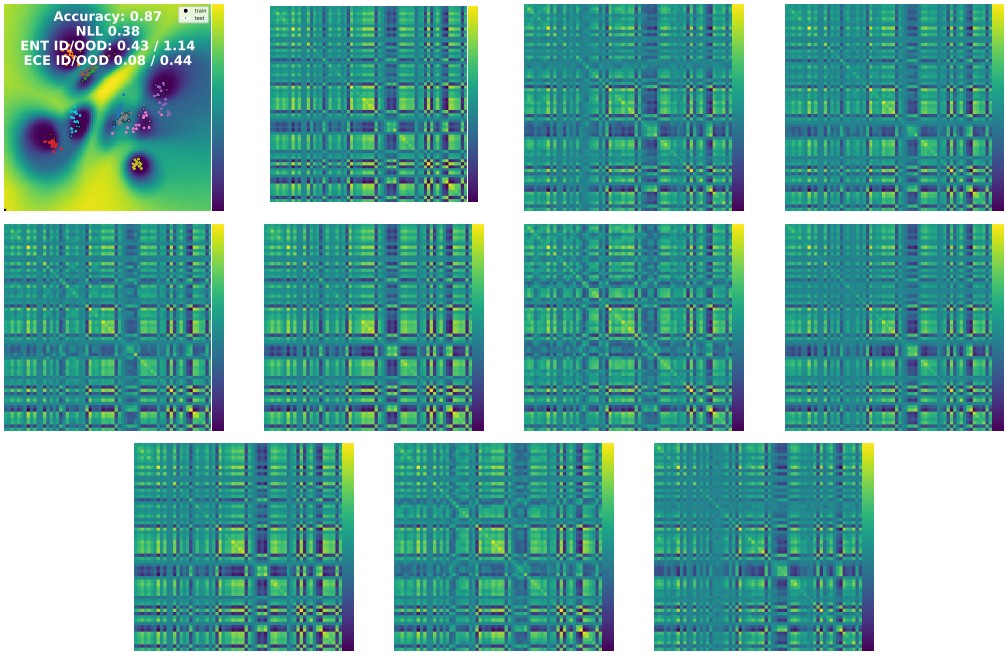

Figure 11: ProtoMahalanobisFC model performance on the meta Gaussians toy dataset. From the top left: Entropy surface, covariances for clases 1-10

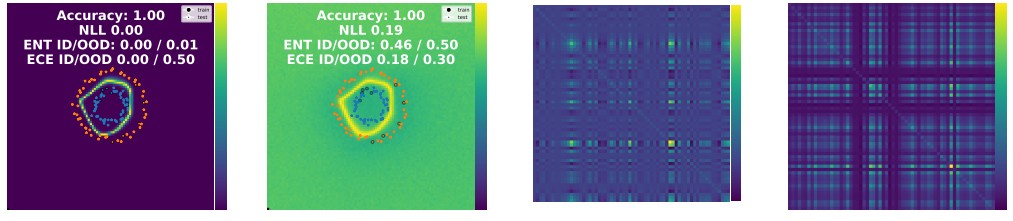

(a) Meta Circles (DDU). From left to right: entropy surface (distance), entropy surface (softmax sample), covariances for class 1-2

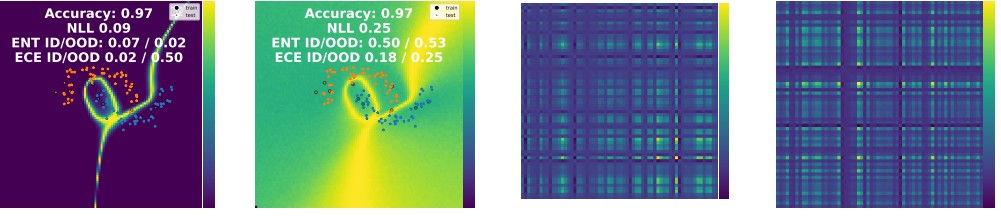

(b) Meta Moons (DDU). From left to right: entropy surface (distance), entropy surface (softmax sample), covariances for class 1-2

Figure 12: Proto DDU model performance on the two toy meta learning datasets.

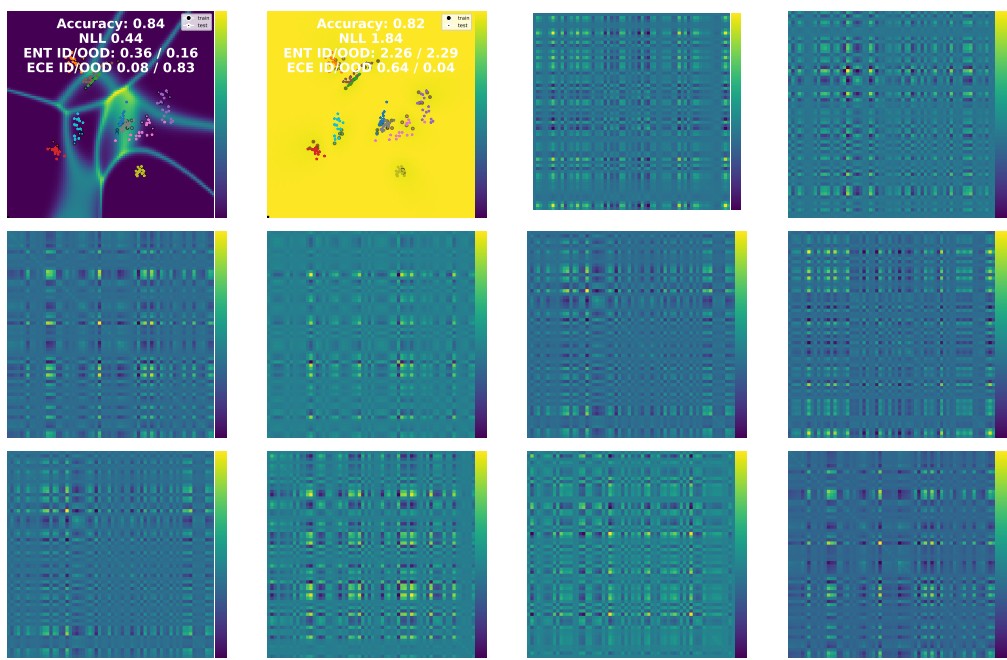

Figure 13: Proto DDU model performance on the Meta Gaussians dataset. From the top left: Entropy surface (distance), entropy surface (softmax sample), covariances for clases 1-10

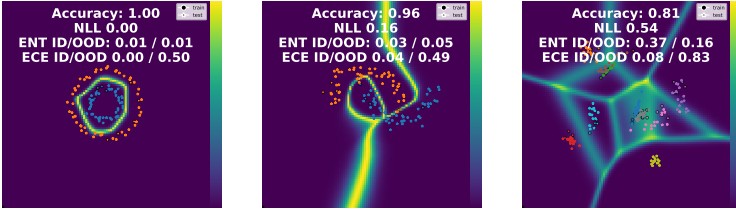

Figure 14: Protonet model performance on the three meta-toy datasets. From left to right: Meta-Circles, Meta-Moons, Meta-Gaussians.



(a) Meta Circles (SNGPProtoFC). From left to right: entropy surface (softmax sample), entropy surface (distance), covariances for class 1-2



(b) Meta Moons (SNGPProtoFC). From left to right: entropy surface (softmax sample), entropy surface (distance), covariances for class 1-2

Figure 15: SNGPProtoFC model performance on the meta-moons and meta-circles toy datasets.

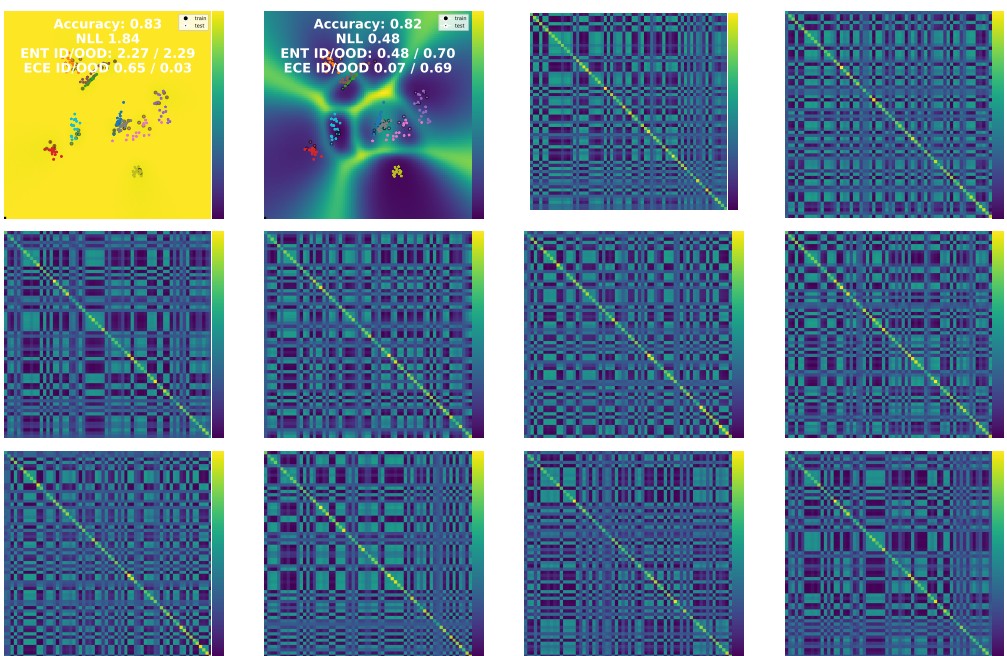

Figure 16: SNGPProtoFC model performance on the meta-Gaussians toy dataset. From the top left: Entropy surface (softmax sample), entropy surface (distance), covariances for clases 1-10

## A.6    FURTHER IMPLEMENTATION DETAILS

**Positive Diagonal Constraint**    In order to constrain the diagonal $\Lambda$ of Proto Mahalanobis models (Equation 5) to be positive as mentioned in Section 3.2, we utilize a truncated sigmoid function $\Lambda = max(0.1, \sigma(z))$. We truncate the values in order to avoid extreme values during the inversion.

**SNGP & DDU**    Both SNGP (Liu et al., 2020a) and DDU (Mukhoti et al., 2021) were originally designed under the assumption that an entire dataset would be used in the final pass to construct a feature covariance matrix. Given that few-shot-learning contains a limited number of samples for each task, we compose the feature covariance as a diagonal + low-rank factor $\Lambda + \Phi\Phi^\top$, where $\Lambda$ is a positive constrained (via softplus) meta learned parameter. $\Lambda$ can be seen as a shrinkage estimation $(\delta\Lambda + (1 - \delta)\Phi\Phi^\top)$ for low sample size, with a meta learned mixing coefficient $\delta$.

In order to extend SNGP to work in the few shot learning scenario under the prototypical network Snell et al. (2017) framework, we had to modify the original algorithm by replacing the last linear layer with the embedding layer and centroids used by prototypical networks. Empirically, we found that using the SNGP logit-normal inference procedure led to a severe performance decrease, therefore our results utilized Mahalanobis distance instead.

**OOD AUPR/AUROC**    In order to evaluate the OOD AUPR/AUROC metrics in the supplementary tables, we utilize the method proposed by Liu et al. (2020b). Specifically, we use the total energy in the logits $\log \sum_i \exp(z_i)$ as the score when evaluating AUPR/AUROC.

**Optimizers**    All models are trained with the Adam (Kingma & Ba, 2014) optimizer

## A.7    ADDITIONAL EIGENVALUE DISTRIBUTIONS

The eigenvalue distributions highlighted in section 4 exhibit the most diverse case of eigenvalues. However, the eigenvalues of ProtoMahalanobis precision matrices become less diverse in the one-shot setting which is also where we are unable to mean center the respective features by class.

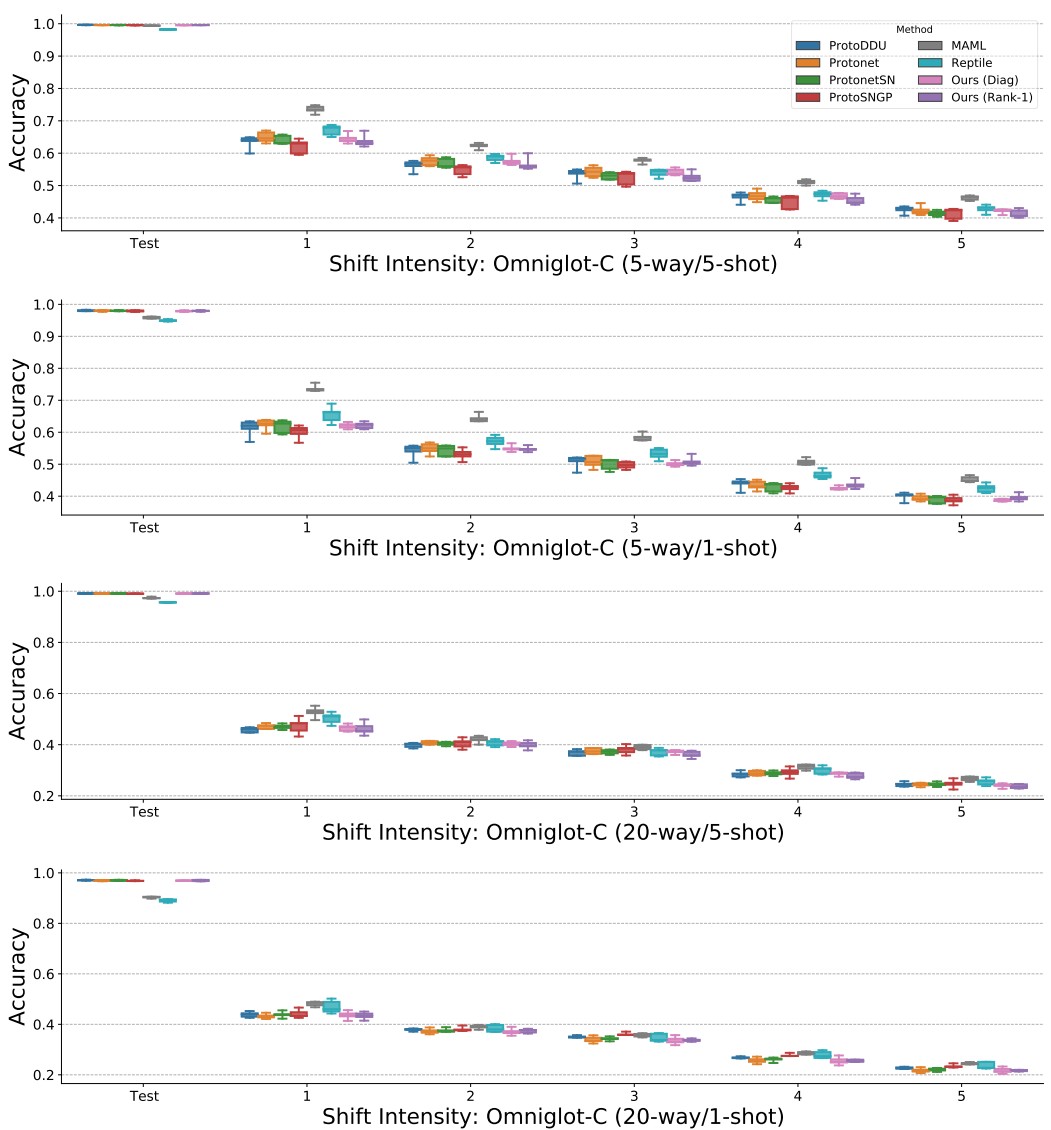

Figure 18: Accuracy boxplots for different variations of the Omniglot dataset

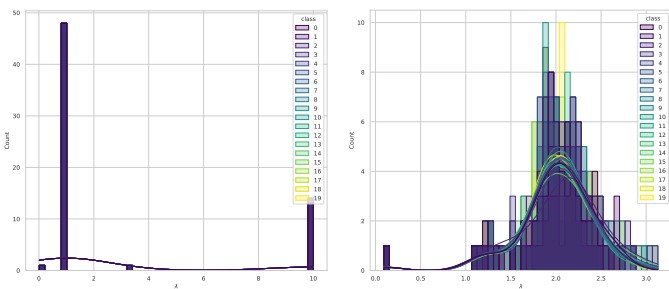

Figure 17: From left to right: covariance, precision, and eigenvalue distribution for ProtoMahalanobis precision matrix on Omniglot 20-way/1-shot (left) and 20-way/5-shot (right) experiments.

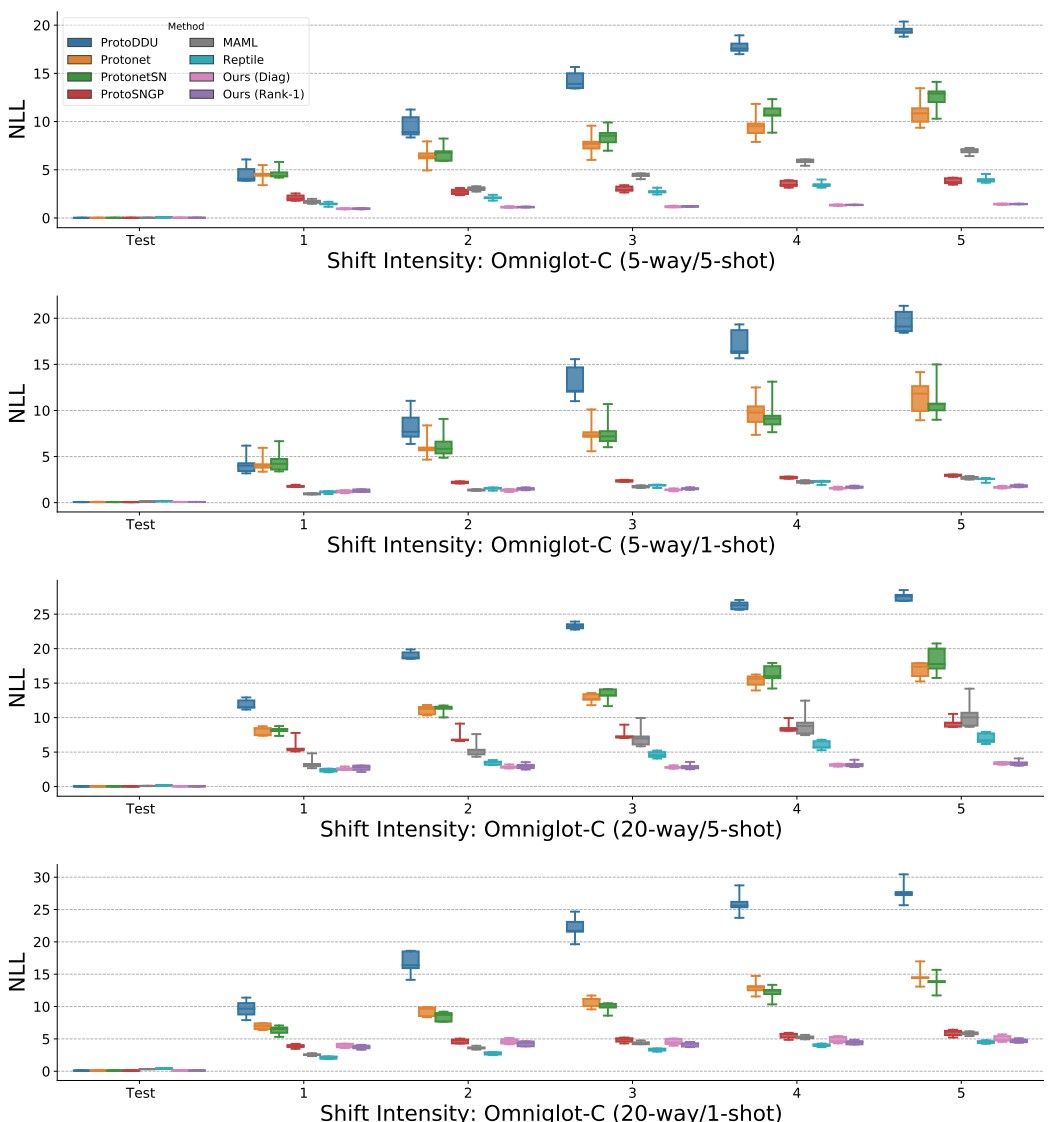

Figure 19: NLL boxplots for different variations of the Omniglot dataset

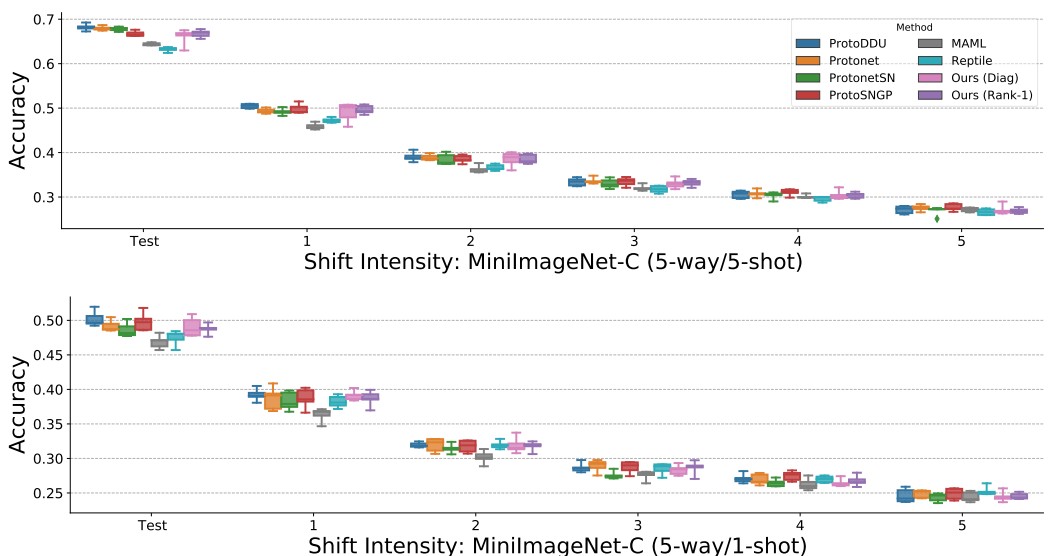

Figure 20: Accuracy boxplots for different variations of the MiniImageNet dataset

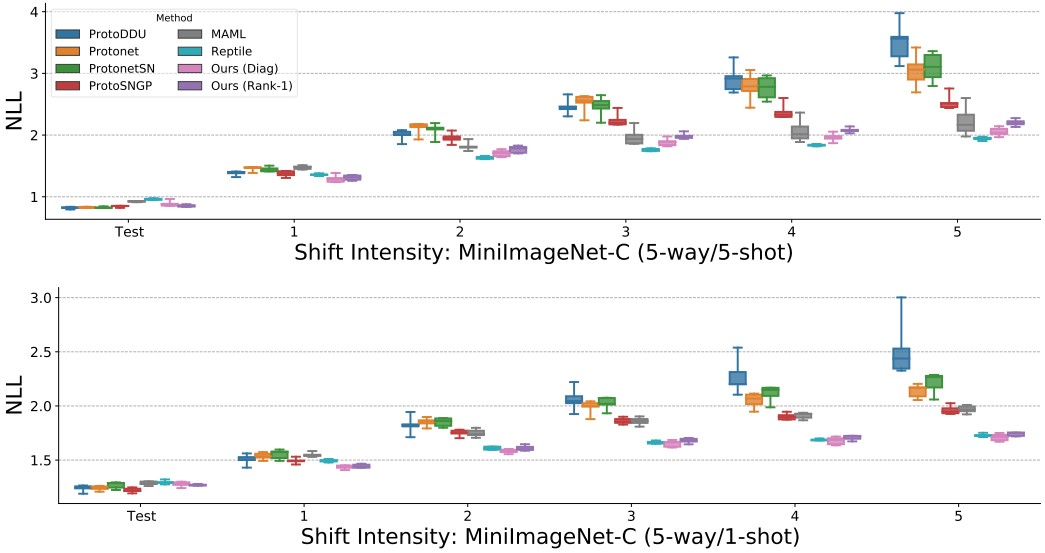

Figure 21: NLL boxplots for different variations of the MiniIMageNet dataset

Table 9: Convolutional architecture used for MAML/Reptile Omniglot

| Layers |
| --- |
| Conv2d(1, 64, pad=1, stride=2) → BatchNorm(reptilenorm=True) → ReLU |
| Conv2d(64, 64, pad=1, stride=2) → BatchNorm(reptilenorm=True) → ReLU |
| Conv2d(64, 64, pad=1, stride=2) → BatchNorm(reptilenorm=True) → ReLU |
| Conv2d(64, 64, pad=1, stride=2) → BatchNorm(reptilenorm=True) → ReLU |
| AveragePool(2) |
| FC(64, nway) |

Table 10: Convolutional architecture used for MAML/Reptile MiniImageNet. Reptile uses 64 filters instead of 32.

| Layers |
| --- |
| Conv2d(1, 32, pad=1, stride=1) → BatchNorm(reptilenorm=True) → ReLU → MaxPool2d(2) |
| Conv2d(32, 32, pad=1, stride=2) → BatchNorm(reptilenorm=True) → ReLU → MaxPool2d(2) |
| Conv2d(32, 32, pad=1, stride=2) → BatchNorm(reptilenorm=True) → ReLU → MaxPool2d(2) |
| Conv2d(32, 32, pad=1, stride=2) → BatchNorm(reptilenorm=True) → ReLU → MaxPool2d(2) |
| Flatten |
| FC(1600, nway) |

## A.8    ADDITIONAL BOXPLOT RESULTS

Figures 18, 20 show extra boxplot results for accuracy while Figures 19, 21 show negative log likelihood.

## A.9    ARCHITECTURE DETAILS

Tables 9, and 10 show the backbone architectures for MAML/Reptile or Omniglot and MiniImageNet respectively. Table 11 shows the backbone architecture for all Protonet based models.

## A.10    RUNTIME ANALYSIS

In Tables 12, and 13 we provide a runtime analysis of different variants of our models and baselines. Linear models are evaluated by using the mean and standard deviations from 50 iterations of both training and inference on the MetaMoons dataset. Convolutional models are likewise evaluated on 50 iterations of the Omniglot dataset. All models were evaluated on a single GeForce GTX 1080 Ti GPU. SNGP/DDU also utilize the matrix inversion outlined in Equation 8.

Mahalanobis models show slightly better (Linear) or similar (CNN) latency to SNGP/DDU for diagonal and rank-1 variants. Latency increases as the rank goes higher due to more factors and more iterations required for inversion and log-determinant calculations. Comparing Protonet, Protonet-SN, and other variants which need to construct a covariance, we can see that constructing the covariance matrix adds a cost which is roughly equivalent to spectral normalization.

Table 11: Convolutional architecture used for Protonet Models. Plain Protonets use no spectral normalization

| Layers |
| --- |
| SpectralNorm(Conv2d(1, 64, pad=1, stride=1), residual=True, c=3) → BatchNorm() → ReLU → Dropout() → AveragePool2d(2) |
| SpectralNorm(Conv2d(1, 64, pad=1, stride=1), residual=True, c=3) → BatchNorm() → ReLU → Dropout() → AveragePool2d(2) |
| SpectralNorm(Conv2d(1, 64, pad=1, stride=1), residual=True, c=3) → BatchNorm() → ReLU → Dropout() → AveragePool2d(2) |
| SpectralNorm(Conv2d(1, 64, pad=1, stride=1), residual=True, c=3) → BatchNorm() → ReLU → Dropout() → AveragePool2d(2) |
| Flatten() |
| FC(features, nway) |

| Model | Train Iteration (ms) | Eval Iteration (ms) |
|---|---|---|
| ProtoMahalanobis-FC diag | 10.33±0.40 | 2.95±0.20 |
| ProtoMahalanobis-FC Rank-1 | 10.96±0.37 | 3.11±0.22 |
| ProtoMahalanobis-FC Rank-5 | 13.06±0.44 | 3.84±0.24 |
| ProtoMahalanobis-FC Rank-10 | 15.65±0.35 | 4.63±0.20 |
| ProtoDDU-FC | 11.86±0.44 | 3.97±0.21 |
| ProtoSNGP-FC | 12.32±0.38 | 4.02±0.22 |
| Protonet-FC | 2.83±0.39 | 0.86±0.09 |
| Protonet-FC SN | 6.78±0.34 | 1.91±0.08 |

Table 12: Runtime analysis of linear variants of models

| Model | Train Iteration (ms) | Eval Iteration (ms) |
|---|---|---|
| ProtoMahalanobis Diag | 11.84±0.69 | 3.61±0.35 |
| ProtoMahalanobis Rank-1 | 12.35±0.75 | 3.85±0.31 |
| ProtoMahalanobis Rank-5 | 15.04±0.69 | 4.55±0.36 |
| ProtoMahalanobis Rank-10 | 17.72±0.71 | 5.42±0.45 |
| ProtoDDU | 11.39±0.81 | 3.80±0.29 |
| ProtoSNGP | 11.43±0.65 | 3.68±0.30 |
| Protonet | 3.55±0.24 | 1.14±0.20 |
| Protonet SN | 8.03±0.40 | 2.42±0.18 |

Table 13: Runtime analysis of CNN variants of models

## A.11 FURTHER EIGENVALUE EXPERIMENTS

In Table 14, we perform further experiments and analysis into the behavior of the low rank covariance encoder outlined in Section 3, we analyze the significance of the eigenvalues of the precision matrix. In this experiment, we first obtain the predicted precision matrix and perform an eigendecomposition $A = Q\Lambda Q^{-1} \in \mathbb{R}^{N \times N}$. We then construct a set of alternate precision matrices $S = \{A'\}_{i=1}^N$, where each set element is a recomposition $A_i' = Q\Lambda_i'Q^{-1}$, where $\Lambda_i'$ has one eigenvalue reset to 1. We then compute the final Accuracy, NLL, and ECE once for each matrix in $S$. If the predicted eigenvalues are due to arbitrary error or noise, then we would expect to see that the test statistics would arbitrarily improve for some precision matrices in $S$.

Instead, in Table 14 we see that the precision matrix which is predicted from the Set Transformer gives the best results on the test set in all cases, showing that all of the predicted values are necessary for the given solution. This experiment utilizes Omniglot 5-way/5-shot and the ProtoMahalanobis Rank-1 variant.

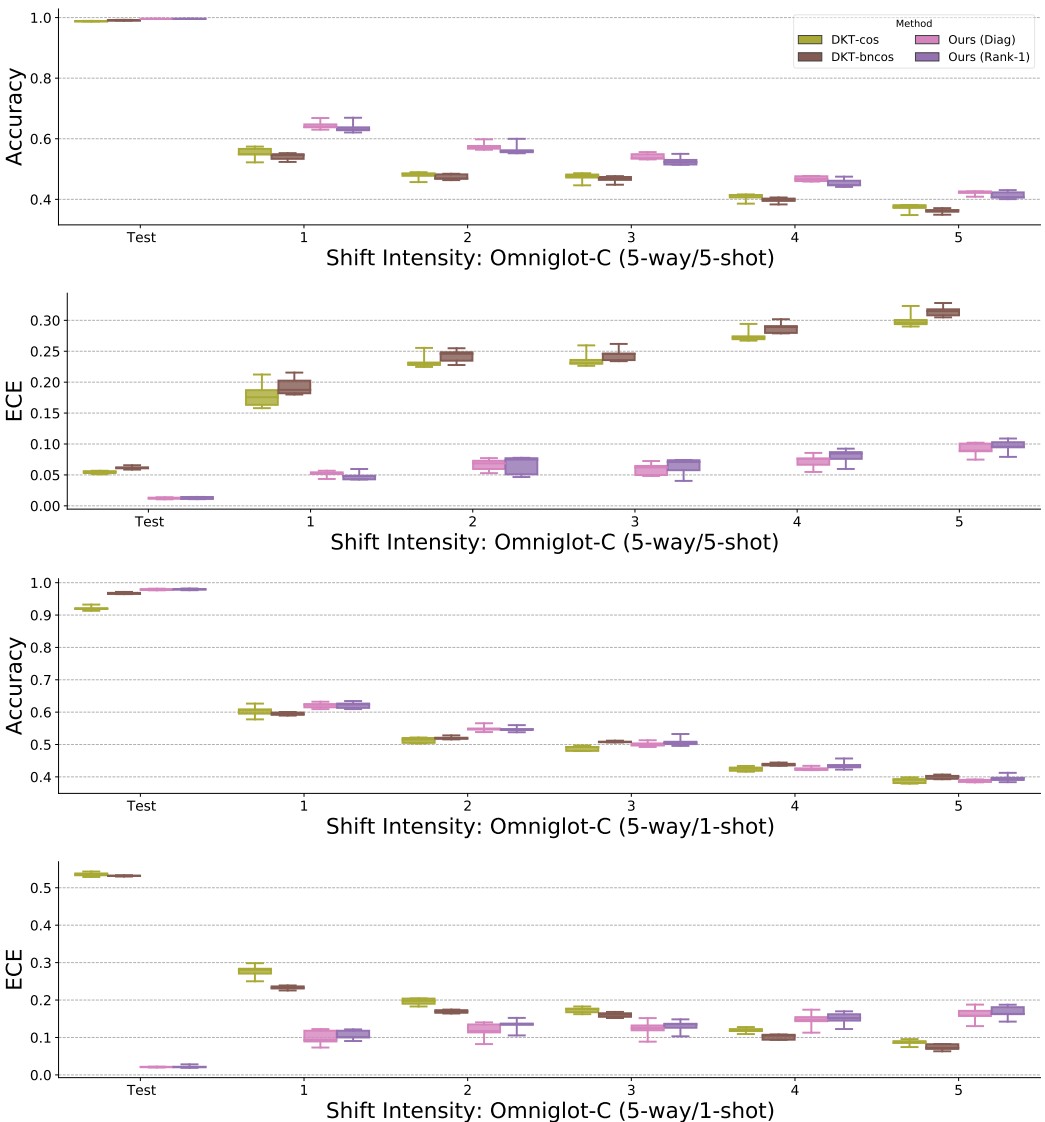

Figure 22: Extra results comparing to Deep Kernel Transfer (Patacchiola et al., 2020) on the Omniglot dataset. In our experiments, DKT showed a large variance in performance between tasks. In the 5-way/1-shot case, calibration on corrupted data comes at the expense of underconfidence on in distribution data.

| $\Sigma^{-1}$ Matrix | Accuracy | NLL | ECE | better% |
|---|---|---|---|---|
| predicted level 0 $\Sigma^{-1}$ | **99.55±0.04** | **0.02±0.00** | **1.21±0.16** | **100%/100%/100%** |
| modified level 0 $\Sigma^{-1}$ | 97.98±0.23 | 0.09±0.01 | 3.31±0.60 | 0%/0%/0% |
| predicted level 1 $\Sigma^{-1}$ | **63.43±1.49** | **0.97±0.05** | **5.22±0.87** | **100%/100%/100%** |
| modified level 1 $\Sigma^{-1}$ | 58.44±2.11 | 1.30±0.17 | 9.06±1.73 | 0%/0%/0% |
| predicted level 2 $\Sigma^{-1}$ | **56.31±1.59** | **1.14±0.04** | **6.92±1.43** | **100%/100%/100%** |
| modified level 2 $\Sigma^{-1}$ | 51.17±2.12 | 1.54±0.37 | 11.92±2.04 | 0%/0%/0% |
| predicted level 3 $\Sigma^{-1}$ | **52.45±1.33** | **1.21±0.04** | **6.62±1.45** | **100%/100%/100%** |
| modified level 3 $\Sigma^{-1}$ | 46.39±1.76 | 1.72±0.74 | 13.51±2.03 | 0%/0%/0% |
| predicted level 4 $\Sigma^{-1}$ | **45.07±1.04** | **1.37±0.02** | **8.34±1.26** | **100%/100%/100%** |
| modified level 4 $\Sigma^{-1}$ | 39.70±1.36 | 1.97±1.13 | 15.85±2.11 | 0%/0%/0% |
| predicted level 5 $\Sigma^{-1}$ | **40.73±0.70** | **1.46±0.02** | **10.25±1.16** | **100%/100%/100%** |
| modified level 5 $\Sigma^{-1}$ | 36.54±0.91 | 2.09±1.42 | 17.34±2.06 | 0%/0%/0% |

Table 14: Analyzing the predicted precision matrix against a set of modified precision matrices with perturbed eigenvalues. The predicted precision matrix performs better in every instance, showing that the precision matrix is not arbitrary. This data comes from Omniglot 5-way/5-shot and utilizes the ProtoMahalanobis Rank-1 variant

