# OpenReview forum: "Meta Learning Low Rank Covariance Factors for Energy Based Deterministic Uncertainty"
_ICLR.cc/2022/Conference — ICLR 2022 Poster_

### Official Review · Reviewer_eieK · 2021-11-02

**Correctness:** 3
**Technical Novelty And Significance:** 3
**Empirical Novelty And Significance:** 3
**Recommendation:** 6
**Confidence:** 3

**Main Review:**

The paper is well written and easy to follow. Some specific comments / suggestions:
1. It would be great if the authors could expand the discussion of bi-Lipschitz regularization in the related work section. Some discussion and literature review of set-input functions, e.g., Set Transformer, Deep Sets will also be helpful.
2. What are the practical ways to ensure that Lambda_c matrix is non-negative, without making too many entries of Lambda_c to be zero?
3. Is there any benchmark on the computational requirement of the proposed methods, compared to competing methods? The covariance inversion and log-determinant will add computational cost to the methods, and it would be good to add some analysis to show the tradeoff between computational cost (i.e., number of iterations) versus performance.
4. In the numerical studies, it seems that the proposal performed comparatively better in the low-way low-shot setting. Is this the correct observation? If so, is there any intuition behind this?

**Summary Of The Paper:**

In this paper, the authors proposed methodologies to estimate the covariance matrix of embedded features under the meta-learning setting. The authors first showed that the empirical covariance matrix is inappropriate in this setting and that due to the shift invariant property of softmax function, ProtoNet could produce overly confident predictions.

The authors instead propose to learn the covariance matrix through a Set Transformer structure, where the covariance matrix can be decomposed into the sum of a diagonal component and a low rank component. The authors used scaled energy and bi-Lipschitz regularization to make the methodology more performant.

Numerical studies are performed to evaluate the calibration performance as well as the eigenvalue distribution of several methods.

**Summary Of The Review:**

The paper is well-written with clear motivations and easy to follow presentations of the methodology. The proposal is relatively novel, and could be a meaningful contribution to the meta-learning community.

---

> ### Author Response · Authors · 2021-11-16
> **Author Response**
>
> Thank you for your time in evaluating our work, we will address your comments in turn below:
>
> **1. Some discussion and literature review of set-input functions, e.g., Set Transformer, Deep Sets would be helpful.**
>
> Thank you for pointing this out. We have **added a discussion** about set encoding related works to the appendix A.4
>
> ---
>
> **2. What are the practical ways to ensure that Lambda_c matrix is non-negative, without making too many entries of Lambda_c to be zero?**
>
> We use a **truncated sigmoid activation function** on the values such that the $\lambda_c = max(0.1, \sigma(z))$, the
> sigmoid is truncated in order to prevent extreme values during inversion. We added details about this into section A.5
> of the appendix.
>
> ---
>
> **3. Is there any benchmark on the computational requirement of the proposed methods, compared to competing methods? The
> covariance inversion and log-determinant will add computational cost to the methods, and it would be good to add some
> analysis.**
>
> Indeed the inversion and log-determinant naive calculations would cause larger overhead to the baselines, but the
> iterative process described in **equations 8 and 9 can be applied directly to the baselines methods as well** since the
> outer product $AA^\top$ can be expressed as a sum of outer products $\sum_i a_i a_i^\top$. As these are exact identities
> and not approximations, we apply equations 8 and 9 to the baselines as well.
>
> As the number of iterations is directly tied to the rank, (or the set size in the baselines), we provide a breakdown
> of the training and inference times here, and in Section A.9 of the appendix. Numbers reported are the average of 50
> iterations on a GeForce GTX 1080 Ti GPU on the MetaMoons (linear variants) and Omniglot (CNN) variants with a batch size
> of 32.
>
> ProtoMahalanobis models are **roughly equivalent** to the ProtoDDU and ProtoSNGP baselines in clock time although the times
> do become slower as the rank increases and more iterations are required.
>
> | Model                     | Train Iteration (ms) | Eval Iteration (ms) |
> |---------------------------|----------------------|---------------------|
> | ProtoMahalanobisFC Diag   |         10.33        |         2.95        |
> | ProtoMahalanobisFC Rank-1 |         10.96        |         3.11        |
> | ProtoMahalanobisFC Rank-5 |         13.06        |         3.84        |
> | ProtoMahalanobisFC Rank-10 |         15.65        |         4.63        |
> | ProtoDDUFC                |         11.86        |         3.97        |
> | ProtoSNGPUFC              |         12.32        |         4.02        |
> | ProtonetFC                |         2.83         |         0.86        |
> | ProtonetFC Spectral       |         6.78         |         1.91        |
>
>
> | Model                   | Train Iteration (ms) | Eval Iteration (ms) |
> |-------------------------|----------------------|---------------------|
> | ProtoMahalanobis Diag   |         11.84        |         3.61        |
> | ProtoMahalanobis Rank-1 |         12.35        |         3.85        |
> | ProtoMahalanobis Rank-5 |         15.04        |         4.55        |
> | ProtoMahalanobis Rank-10 |         17.72        |         5.42        |
> | ProtoDDU                |         11.39        |         3.80        |
> | ProtoSNGPU              |         11.43        |         3.68        |
> | Protonet                |         3.55         |         1.14        |
> | Protonet Spectral       |         8.03         |         2.42        |
>
> ---
>
> **4. In the numerical studies, it seems that the proposal performed comparatively better in the low-way low-shot setting.
> Is this the correct observation? If so, is there any intuition behind this?**
>
> There are a few things going on here. Firstly, between 1-shot and 5-shot experiments of ProtoMahalanobis, we would
> expect 5-shot to perform better as there are more examples which can be encoded into low rank covariance factors. We do
> observe this trend. Secondly, we observe a decrease in absolute performance in the higher way experiments, as these are
> generally harder problems.
>
> ---
>
> Thank you again for your time and effort in evaluating our work, we remain open to answering any remaining questions you have.

---

### Official Review · Reviewer_QrCN · 2021-11-02

**Correctness:** 3
**Technical Novelty And Significance:** 2
**Empirical Novelty And Significance:** 2
**Recommendation:** 5
**Confidence:** 2

**Main Review:**

This submission is empirical: the paper is well-motivated, and the paper adapts existing, popular techniques to engineer a solution to the standing problem (using quadratic discriminant analysis, modeling the covariance matrix as a sum of a diagonal matrix and a low-rank psd matrix, etc.).

On the downside, there are several theoretical questions that are left unanswered:
1. Why is the Set Transformer a good choice? On which ground can we justify that the Set Transformer can generate sensible values of Lambda and Phi to construct the covariance matrices?
2. There are a lot of methods from statistics to estimate covariance matrices from low sample size dataset (linear/nonlinear shrinkage, sparse, etc.). Why can't they be used in this application? Is it possible to show that the Set Transformer is better than these traditional approach (for any possible criteria)?
3. In page 8, why should the eigenvalue distribution be diversified? It is well known that the eigenvalue can be sensitive to small perturbations of the off-diagonal entries of the matrix. Thus, what we observed (diversity of the eigenvalues) can be just an artifact of the numerical errors from the Set transformer network. Diverse eigenvalues also imply that the condition number is large, which leads to higher numerical errors to the log-determinant term. Thus, I am not fully convinced that a diverse set of eigenvalues is of any advantage.


Minor:
- What is f(E) and min(E) after equation (10)?
- The abstract and intro focus heavily on bi-Lipschitz regularized NN, however, the exposition in section 3 (except for Section 3.4 which is 10 lines long) relies minimally on this architecture. This emphasis may be misleading in the first read.

**Summary Of The Paper:**

This submission focus on improving the performance of uncertainty quantification and OOD detection using bi-Lipschitz regularized neural networks. There are two involved challenges here: (i) we are required to estimate the class-conditional covariance matrix of the latent features. This estimation is challenging under low sample size. (ii) generative classifiers using softmax activation function suffer from shift invariance, and thus may become over-confident for OOD samples.

To resolve these issues, the paper focus on two innovations:

(i) a meta-learning model to estimate the class-conditional covariance matrices,
(ii) a modification of the inference procedure with the energy function to detect OOD samples.

The paper concludes with numerical results showing the performance of the proposed solution approach.

**Summary Of The Review:**

In my opinion, the paper is a good engineering attempt to solve a specific problem. It lacks, unfortunately, concrete scientific contributions.


======== Post-rebuttal ========
The authors have provided further empirical results to justify the use of the ST to estimate the covariance matrix. I have raised my evaluation from 3 to 5.

---

> ### Author Response · Authors · 2021-11-16
> **Author Response (Part 2)**
>
> **3. In page 8, why should the eigenvalue distribution be diversified? It is well known that the eigenvalue can be
> sensitive to small off-diagonal perturbations. Thus, diversity of the eigenvalues can be an artifact of the numerical
> errors from the ST.**
>
> - Proto SNGP/DDU both learn a covariance which has most eigenvalues close to 1. This indicates that the learned Gaussian
> is **near spherical and identical** for every class. ProtoMahalanobis, on the other hand, shows a diverse range of
> eigenvalues which vary by class indicating a non-trivial, class-dependent distribution.
>
> - In order to further examine the effect of the eigenvalues of the predicted precision matrix, we conducted an experiment
> where we obtain the predicted precision matrix from the ST and perform an eigendecomposition $A = Q \Lambda Q^{-1} \in
> \mathbb{R}^{N \times N}$. We then construct a set of alternate precision matrices $S = \\{A_i'\\}_{i=1}^N$, where each
> element is a recomposition $Q \Lambda'Q^{-1}$, where $\Lambda'$ has one eigenvalue reset to 1. We then compute the
> final Accuracy, NLL, and ECE once for each precision matrix in $S$. **If the predicted eigenvalues of the precision matrix
> are due to arbitrary error/noise**, then we would **expect to see that the test statistics would arbitrarily improve** for
> some precision matrices in $S$.
>
> - This experiment was done in the same setting as figure 6 of the text. 'Pert. Precision' is the average prediction from
>   all 64 perturbed matrices in $S$. The rightmost column shows the percentage of times that a particular precision matrix gave better
>   respective Accuracy, NLL, or ECE on corruption levels 0-5.
>
> - We find that the predicted precision matrix from the ST gives the **best results on the test set in all cases**, showing
>   that the eigenvalue distribution is **not the result of errors or noise from the ST.** We have added this experiment to
>   section A.10 in the appendix (Table 14).
>
> | Precision Matrix Type     | Accuracy |  NLL |  ECE  |   Better (%)   |
> |---------------------------|:--------:|:----:|:-----:|:--------------:|
> | ST Precision (level 0)    |   **99.55**  | **0.02** |  **1.21** | **100%/100%/100%** |
> | Pert. Precision (level 0) |   97.98  | 0.09 |  3.31 |    0%/0%/0%    |
> | ST Precision (level 1)    |   **63.43**  | **0.97** |  **5.22** | **100%/100%/100%** |
> | Pert. Precision (level 1) |   58.44  | 1.30 |  9.06 |    0%/0%/0%    |
> | ST Precision (level 2)    |   **56.31**  | **1.14** |  **6.92** | **100%/100%/100%** |
> | Pert. Precision (level 2) |   51.17  | 1.54 | 11.92 |    0%/0%/0%    |
> | ST Precision (level 3)    |   **52.45**  | **1.21** |  **6.62** | **100%/100%/100%** |
> | Pert. Precision (level 3) |   46.39  | 1.72 | 13.51 |    0%/0%/0%    |
> | ST Precision (level 4)    |   **45.07**  | **1.37** |  **8.34** | **100%/100%/100%** |
> | Pert. Precision (level 4) |   39.70  | 1.97 | 15.85 |    0%/0%/0%    |
> | ST Precision (level 5)    |   **40.73**  | **1.46** | **10.25** | **100%/100%/100%** |
> | Pert. Precision (level 5) |   36.54  | 2.09 | 17.34 |    0%/0%/0%    |
>
> ---
>
> ### Minor
>
> **What is f(E) and min(E) after equation (10)?**
>
> $f(E)$ is the same function as $h(E)$ introduced in the beginning of section 3.3. The minimum should be over the class
> specific energies $min_c(E)$, we have fixed this typo.
>
> **The abstract and intro focus heavily on bi-Lipschitz regularized NN, however, the exposition in section 3 (except for
> Section 3.4 which is 10 lines long) relies minimally on this architecture. This emphasis may be misleading in the first
> read.**
>
> The bi-Lipschitz extractor is heavily justified in previous works (which we used for both Proto-SNGP and Proto-DDU
> baselines) and gives our model the necessary inductive bias needed to work. An even earlier work, which is the first
> work we know of that used this regularization regarding uncertainty is [1]. As we highlight in section 2, bi-Lipschitz
> regularization prevents feature collapse and allows for meaningful distances to be preserved in the latent space. We
> re-iterate in section 3.4, that we directly utilize distance, requiring the bi-Lipschitz regularization.
>
> ---
>
> **In my opinion, the paper is a good engineering attempt to solve a specific problem. It lacks, unfortunately, concrete
> scientific contributions.**
>
> As stated at the end of the introduction, **our main contributions are**:
>
> 1. The proposal of a covariance encoder which encodes inputs sets into low rank covariance factors.
>
> 2. We highlight the problem of softmax shift invariance as it relates to calibration in centroid classifiers
>    (illustrated in figure 3), and the proposal of our inference procedure outlined in equations 10-12 which mitigates
>    this problem.
>
> We have also provided ample evidence that our solution is effective and achieved a lower calibration error on varying
> levels of OOD corruptions.
>
> Thank you again for your time and effort in evaluating our work, we remain open to answering any questions which may
> remain.

---

> > ### Comment · Reviewer_QrCN · 2021-12-01
> > **Thanks for the response... but still some rigorous justifications are missing**
> >
> > I would like to thank the authors for their replies.
> >
> > As I have specified in my original review, I am satisfied with the empirical results of this paper. What I am asking is about the theoretical justifications of such an approach. The authors' replies focus, nevertheless, on further experimental results. These extra numerical results provide further confidence for using the method, but are also redundant because they do not shed any insights on the true mechanisms underlying this approach.
> >
> > In my opinion, using a neural network (ST) to estimate a well-known statistical quantity (covariance matrix) under low sample size (few-shot learning) is inefficient. It is mysterious, and may be an open question to explore next, which signal are being captured by ST and how this signal can improve the performance (as shown by the empirical results).
> >
> > I'm happy to raise my evaluation to a 5. I'm sorry for the late reply.

---

> > > ### Author Response · Authors · 2021-12-01
> > > **Thank you for the response**
> > >
> > > Thank you for the response. We would like to kindly point out that the extra results we provided directly address the concerns which were raised in the initial review.
> > >
> > > As stated, this is an empirical work highlighting a problem when applying existing approaches to the meta learning setting. As such, we disagree that theoretical justification is necessary for our work to be relevant to the community.
> > >
> > > Thank you for your continued evaluation,
> > >
> > > Authors

---

> > > > ### Comment · Reviewer_QrCN · 2021-12-06
> > > > **I do not agree with the authors**
> > > >
> > > > Unfortunately, I need to say that I do not necessarily agree with the authors' reply. I believe that an important part of scientific (academic) research is to study how/why/when a particular method works or does not work.
> > > >
> > > > Even an empirical paper should (at least) contain the following two components:
> > > > 1. What are the desiderata of the solution?
> > > > 2. Justifications on why the chosen method will help achieving the desiderata in point 1.
> > > >
> > > > Holistically speaking, investigating the above two points has lead the community to many ingenious solutions (for example, convolutional neural net, LSTM, attention, etc.) Those are truly the methods that are "relevant to the community".
> > > >
> > > > Getting back to this paper, can the authors please answer the 2 points above? Throughout the paper, I could not find a desiderata for a good covariance matrix estimator that can perform well in the setting of interest. Using ST is just adding another layer of ``black box", and does not necessarily answer point 2.
> > > >
> > > > Why am I asking this? Because the statistics community have been doing this practice for a long time. For example, the community know the Marcenko-Pastur law for the distribution of the eigenvalue, the community know the Jensen inequality leading to the over-estimation of the largest empirical eigenvalue and to the under-estimation of the smallest empirical eigenvalue (which in turns imply that the eigenvalues of the sample covariance matrix is too much spread-out and require shrinkage), the community know from numerical stability that the condition number should be as small as possible, etc... Those are the desiderata, which lead to many methods for better estimating the covariance matrix (a google research returns a few hundreds of papers using a combination of these desiderata)...
> > > >
> > > > I thus have to say that a 5 is the maximum score that I can give for your paper as of now. As I have pointed out from the first round, the paper lacks scientific understanding. I am very eager to increase my score, even be supportive, if the above 2 points are well addressed (what effects do we need and why ST achieve these effects?). Please refrain from answering generally, e.g., ST will meta-learn while other methods do not, because these answers are uninformative.

---

> > > > > ### Author Response · Authors · 2021-12-07
> > > > > **Learning from the task distribution is the desiderata**
> > > > >
> > > > > Thank you for your response.
> > > > >
> > > > > In this paper, we focus on developing a data-driven approach to solve the target problem. Therefore, our desiderata is the ability to learn from the task distribution.
> > > > >
> > > > > We can categorize machine learning approaches into two directions. The first one, as you mentioned, is to introduce carefully curated inductive biases to fit the specific task, something the statistics community has been doing for a long time. On the other hand, data-driven approaches represent a huge line of research, aiming to draw algorithms and methods from data itself. The key is how to effectively allow this to happen. A well known example is, unarguably, meta-learning, where we assume learning from a task distribution as the de-facto desiderata.
> > > > >
> > > > > Then, what architectures/methods should be used to satisfy the above desiderata? The answer is simple. Any architectures/methods can be used as long as they can flexibly learn from the task distribution, and thereby generalize to related, but unseen tasks. This is how meta-learning approaches are justified.
> > > > >
> > > > > A canonical example is Neural Processes [1]. In NP’s, the inference network is amortized over the task distribution. Beyond satisfying some base conditions (exchangeability and consistency in the case of NP’s), no further justification is required. As long as the inference network can empirically learn the task distribution without overfitting, it is sufficient.
> > > > >
> > > > > Another example is the learned optimizer [2], which introduces a learnable LSTM optimizer, amortized across all timesteps of the learning processes. How should the LSTM be theoretically justified over a transformer or any other sequential model? The fact is, LSTMs contain the proper base conditions (sequential model) to perform the specified job, and are justified by empirical success, no further justification required. [2] proposes a new data driven solution to a problem, which is both their contribution and justification.
> > > > >
> > > > > Likewise, in our work, we specify the ST as a covariance predictor in a new way such that it satisfies the base conditions (generating low rank PSD covariance, inversion, pairwise interactions). In generating class-conditional covariance matrices, we empirically verified that our formulation of the ST is able to:
> > > > >
> > > > > - flexibly amortize the class-conditional covariance factors over the task distribution
> > > > > - generalize to unseen tasks, outperforming baselines using traditional methods.
> > > > > - predict covariance factors which are not arbitrary.
> > > > >
> > > > > In this way, the use of ST is justified by both fulfilling the base conditions, and empirical performance on the desiderata.
> > > > >
> > > > > Again, we thank you for your discussion, and please let us know if you have any further concerns.
> > > > >
> > > > > Authors
> > > > >
> > > > > ## References
> > > > >
> > > > > [1] Garnelo et al., Neural Processes, 2018.
> > > > >
> > > > > [2] Andrychowicz et al., Learning to learn by gradient descent by gradient descent, NIPS 2016.

---

> ### Author Response · Authors · 2021-11-16
> **Author Response (Part 1)**
>
> Thank you for your time and effort in reviewing our work, we will address your comments in turn below:
>
> **1. Why is the Set Transformer (ST) a good choice? How to justify that the ST can generate sensible values?**
>
> - The ST is justified because it is **meta learned** across a distribution of tasks to minimize the query loss.
>   Architecturally, The ST is a good choice because it considers **pairwise interactions between set elements**
>   (transformer), which is a useful inductive bias for constructing covariance factors.
> - For example, the calculation of covariance $\boldsymbol{\Sigma} = (\boldsymbol{X} -
>   \mu(\boldsymbol{X}))(\boldsymbol{X} - \mu(\boldsymbol{X}))^\top$ has a **similar structure as the query key
>   multiplication within the self-attention** of the transformer $\sigma(QK^\top)$
> - This choice is affirmed in both our **empirical results**, which outperform the baselines, and is **further demonstrated in
>   figure 6** and the **additional experiment below** (which is also Table 14 of the updated text), which shows that
>   nontrivial, nonrandom predictions come from the ST. We have also added this justification to our discussion of set
>   encoders in the appendix A.4
>
> ---
>
> **2. There are many statistics methods to estimate cov. matrices from low sample size (linear/nonlinear shrinkage,
> sparse, etc.). Why can't they be used? Is it possible to show that the Set Transformer outperforms these?**
>
> - In our baselines, we have chosen methods which construct covariances from meta learned feature extractors. The use of
>   the set transformer (ST), in conjunction with our experimental results, shows that the ST is **meta learning information
>   which is not learned by a meta learned feature extractor.**
>
> - The protonet baselines represents a fixed, identity covariance matrix as we outlined in section 3.1
>
> - Shrinkage can be classified as a ridge-type estimator [2], and **we are actually using this in baseline models** (ProtoDDU,
>   ProtoSNGP) which utilize a meta trainable parameter as a diagonal $\boldsymbol{\Lambda}$ for the covariance
>   $\boldsymbol{\Lambda} + \boldsymbol{\Phi} \boldsymbol{\Phi}^\top$. This can be seen as using a **meta learned shrinkage**
>   parameter $\delta$ in the form of $(\delta \boldsymbol{\Lambda} + (1 - \delta) \boldsymbol{\Phi \Phi^\top})$. The meta
>   learned $\boldsymbol{\Lambda}$ was necessary to stabilize the training process with smaller task sizes.  We have
>   added this implementation detail to the appendix A.6
>
> - The baselines are also meta learning over the task distribution which means that they don't necessarily have a problem
>   with sample size, as it can also be a **baseline failure to encode meta information** about the covariance over the task
>   distribution.
>
> - Additionally, we perform extra experiments trying to increase the meta learned information and degrees of freedom in
>   the baselines by doubling the size of the embedding vector and subsequently using a **dedicated section only for
>   covariance factors $\boldsymbol{\Phi}$.** We provide these results below (the new model is DDU 2p) which show no meaningful improvement in baseline performance.
>
> |               |  Omniglot  |            |             |             | MiniImageNet |            |
> |---------------|:----------:|:----------:|:-----------:|:-----------:|:------------:|:----------:|
> | model         | 5way/5shot | 5way/1shot | 20way/5shot | 20way/1shot |  5way/1shot  | 5way/5shot |
> | DDU           |    69.16   |    66.61   |    78.14    |    71.39    |     35.31    |    46.82   |
> | DDU (2p)      |    69.18   |    66.68   |    78.64    |    72.66    |     33.25    |    46.90   |
> | Ours (Diag)   |    33.95   |    40.52   |    **40.00**    |    50.39    |     **17.34**    |    **32.27**   |
> | Ours (Rank-1) |    **33.19**   |    **39.62**   |    40.04    |    **49.28**    |     18.86    |    34.47   |

---

> ### Author Response · Authors · 2021-11-22
> **Please see our further experimental results**
>
> Dear QrCN,
>
> We have done our best to address each of your concerns. We have:
>
> 1. Justified the use of the set transformer through the similarities of the covariance and dot-product attention calculations.
> 2. Highlighted that we are already using a meta learned form of shrinkage in the baselines which aids in the low sample regime.
> 3. Provided further experimental results showing that the precision matrices which result from the set transformer are not trivial, not the result of errors, and that perturbing the eigenvalues in any dimension results in lower performance in all cases.
> 4. Provided an additional experiment, attempting to boost the baseline performance by providing more parameters
> and more dedicated dimensions for constructing covariances which also failed to improve baseline performance.
>
> The discussion period is coming to a close soon, and we would like to take the time to address any concerns which may
> remain after evaluating our responses.
>
> Thank you for your time and effort in reviewing our work,
>
> Authors

---

> ### Author Response · Authors · 2021-11-29
> **Discussion is coming to a close**
>
> Dear QrCN,
>
> The discussion period is coming to a close within a day, and we will not be able to directly engage in further discussions after that point. We have done our best to address the questions raised in your review, and we remain open to discussing any remaining concerns you may have until the very end. If there are any remaining questions or concerns, please do not hesitate to ask.
>
> Thank you for taking the time to evaluate our work and our responses,
>
> Authors

---

### Official Review · Reviewer_Zz5v · 2021-11-03

**Correctness:** 4
**Technical Novelty And Significance:** 3
**Empirical Novelty And Significance:** 3
**Recommendation:** 6
**Confidence:** 4

**Main Review:**

Strengths:
The overall presentation of this paper is both clear and straightforward.

The figures and equations precisely facilitate the reading.

No noticeable typos were spotted.

The topic that the authors selected is important and largely overlooked by the research of meta-learning and few-shot learning. The effectiveness of the proposed method is reflected by the quantitative comparisons.

Weakness:
My major concern to this paper mainly arises from the use of an additional meta-learned set encoder to predict the task-specific covariance.
It has been shown in previous work that some meta-learners have natural advantages in quantifying uncertainty.

For example, relying on kernel methods, Gaussian processes based meta-learners [1,2] have demonstrated strong uncertainty calibration without leveraging any auxiliary components.

Additionally, I might be wrong, but I'm curious that if this class-level covariance can be obtained through aggregating sample-wise information so that an additional set encoder can be removed.
Please refer to Eq.6 to Eq. 9 of [3] for an example. Although the motivations are different, I believe there is some shared wisdom.

[1] Deep kernel transfer in gaussian processes for few-shot learning, NeurIPS 2020

[2] Bayesian Few-Shot Classiﬁcation with One-vs-Each Pólya-Gamma Augmented Gaussian Processes, ICLR 2021

[3] Amortized Bayesian Prototype Meta-learning: A New Probabilistic Meta-learning Approach to Few-shot Image Classification, AISTAT 2021

**Summary Of The Paper:**

The authors explore the uncertainty calibration problem in meta-learning few-shot classification, which is a relatively new topic that was seldom discussed before.
To solve the challenges that the few samples in few-shot learning are usually insufficient for providing reliable estimation to the covariance matrices, the authors introduce meta-learning covariance matrices parametrized by a set encoder (implemented as set-transformer). And the authors further introduce scaled energy to parameterize a logit-normal softmax distribution for improving the uncertainty calibration of the softmax classification layer.

**Summary Of The Review:**

Overall this paper attacks a very important problem of meta learning and few-shot learning. And considering the quantitative results, the proposed method does achieve better uncertainty calibration.
I expect further discussion on justifying the necessity of adopting an additional set encoder to achieve this capability.

---

> ### Author Response · Authors · 2021-11-16
> **Author Response**
>
> Thank you for your time and effort in evaluating our work, we will address your comments below:
>
> **I might be wrong, but I'm curious that if this class-level covariance can be obtained through aggregating sample-wise
> information so that an additional set encoder can be removed. Please refer to Eq.6 to Eq. 9 of [3] for an example.
> Although the motivations are different, I believe there is some shared wisdom.**
>
> We looked through the mentioned reference [3] and found that the most appropriate aspect of the model would likely be
> the fact that the embedding network embeds the latent features into two sections each responsible for the mean and
> covariance separately. It could be the case that our baselines couldn't encode all of the information about the centroid
> and covariance into the same vector. We tried encoding the mean and covariance factors into separate sections with a
> dedicated section for $\boldsymbol{\Phi}$ using the DDU baselines and achieved the following results.
>
> We ran this experiment with the same settings as Table 1 of the text. **We saw no meaningful improvement in the baseline
> performance** (DDU 2P contains the separate latent dimensions for mean and covariance used in [3]). Additionally, reviewer QrCN
> pointed out additional **sample-wise methods such as shrinkage which were actually used in our baselines**. We have added a
> discussion about shrinkage to appendix A.6
>
> |               |  Omniglot  |            |             |             | MiniImageNet |            |
> |---------------|:----------:|:----------:|:-----------:|:-----------:|:------------:|:----------:|
> | model         | 5way/5shot | 5way/1shot | 20way/5shot | 20way/1shot |  5way/1shot  | 5way/5shot |
> | DDU           |    69.16   |    66.61   |    78.14    |    71.39    |     35.31    |    46.82   |
> | DDU (2p)      |    69.18   |    66.68   |    78.64    |    72.66    |     33.25    |    46.90   |
> | Ours (Diag)   |    33.95   |    40.52   |    **40.00**    |    50.39    |     **17.34**    |    **32.27**   |
> | Ours (Rank-1) |    **33.19**   |    **39.62**   |    40.04    |    **49.28**    |     18.86    |    34.47   |
>
> The main problem with using sample-wise information in this case is that sample-wise information depends on the sample
> (i.e., the current task) and somewhat ignores the fact that we need to **meta learn from the large sample (task
> distribution) which incorporates meta information into the small sample (the task).** The set transformer was used to
> fill this role, where baseline models failed to perform adequately.
>
> Thank you for evaluating our work, we remain open to answering any remaining questions you may have.

---

> ### Author Response · Authors · 2021-11-23
> **New Baseline Added**
>
> Dear Zz5v,
>
> We were able to evaluate a baseline you suggested [1] (DKT) in the same setting as we evaluated our models. In the updated
> text we have replaced figures 4, 5, 19, 20, 21, and 22 to include results with DKT. We provide
> a representative example below (MiniImageNet 5-way/1-shot), and all of the the above quoted figures follow this same pattern.
> We used the original code provided by the author and temperature scaled the trained model as we did for all models.
>
> We found that DKT is significantly underconfident on the natural test data (extremely so in the omniglot 20-way
> experiment in the updated figure 4). While this leads to lower ECE numbers on the higher corruption levels, this is only
> due to the prediction being underconfident and miscalibrated at the lower corruption levels. We suspect the likely culprit of this is the
> one-vs-all binary classification training scheme for each class of DKT which is different (by the original code) than the
> softmax which is necessarily used in the ECE calculation.
>
> On the other hand, our method shows a low ECE on the natural data, which only gradually increases with the corruption level, leading to a more reliable overall confidence at any given level of OOD data.
>
> ### ECE (MiniImageNet 5-way/1-shot)
>
> | Model         | Level 0 | Level 1 | Level 2 | Level 3 | Level 4 | Level 5 |
> |---------------|---------|---------|---------|---------|---------|---------|
> | DKT           | 0.205   | 0.093   | 0.037   | 0.013   | 0.012   | 0.013   |
> | Ours (Diag)   | 0.084   | 0.021   | 0.056   | 0.084   | 0.098   | 0.108   |
> | Ours (Rank-1) | 0.067   | 0.026   | 0.081   | 0.107   | 0.114   | 0.125   |
>
>
> Thank you for the suggestions and your continued evaluation of our work, we remain open to answer any remaining questions
> which you may have.
>
>
> ## References
>
> [1] Patacchiola, M., Turner, J., Crowley, E. J., O'Boyle, M., & Storkey, A. (2020). Bayesian meta-learning for the few-shot setting via deep kernels.

---

### Decision · Program_Chairs · 2022-01-20

**Decision:**

Accept (Poster)

**Comment:**

Reviewers all found the work well-motivated in addressing uncertainty, a topic that has not seen much focus in meta-learning and few-shot learning. They describe the challenges well: small sample sizes and OOD shift. They then propose a solution they find works well empirically to overcome these challenges based on a set encoder and an energy function respectively.

The proposal is largely one of engineering components that have been found to work well in the literature. I'm sympathetic to this style of research (particularly in today's neural network research), although the reviewers raise a primary concern about whether the choices leading to the proposal are justified. In particular, two Reviewers argue that there are no clear ablations compared to alternative simpler approaches, and so the approach of selecting a Set Transformer is rather arbitrary. My perspective is that theory provides one sufficient but not necessary angle to do this, and I do find the authors' replies to the two reviewers convincing. In particular, they add a baseline to estimate covariances suggested by Reviewer Zz5v and they describe how the current baselines do in fact use the shrinkage suggestion by Reviewer QrCN.

I recommend the authors use the reviewers' feedback to enhance their submission's clarity and overall quality.